



**Quantile Sampling: a robust and simplified pixel-based**
**multiple-point simulation approach**
Mathieu Gravey[1], Grégoire Mariethoz[1]
[1] University of Lausanne, Faculty of Geosciences and Environment, Institute of Earth Surface Dynamics,
Switzerland
*Correspondence to*: Mathieu Gravey (mathieu.gravey@unil.ch)
**Highlights**
• A new approach is proposed for pixel-based multiple-point geostatistics simulation.
• The method is flexible and straightforward to parametrize.
• It natively handles continuous and multivariate simulations.
• High computational performance with predictable simulation times.
• A free and open-source implementation is provided.
**Abstract**
Multiple-point geostatistics enable the realistic simulation of complex spatial structures by
inferring statistics from a training image. These methods are typically computationally
expensive and require complex algorithmic parametrizations. The approach that is presented in
this paper is easier to use than existing algorithms, as it requires few independent algorithmic
parameters. It is natively designed for handling continuous variables, and quickly implemented
by capitalizing on standard libraries. The algorithm can handle incomplete training images of
any dimensionality, with categorical or/and continuous variables, and stationarity is not
explicitly required. It is possible to perform unconditional or conditional simulations, even with
exhaustively informed covariates. The method provides new degrees of freedom by allowing
kernel weighting for pattern matching. Computationally, it is adapted to modern architectures
and runs in constant time. The approach is benchmarked against a state-of-the-art method. An
efficient open-source implementation of the algorithm is released and can be found here
(https://github.com/GAIA-UNIL/G2S), to promote reuse and further evolution.
**Keywords**
Multiple-point statistics, stochastic simulation, continuous variable, training image, cross-
correlation, Fourier transform.
**1. Introduction**
Geostatistics is widely used to generate stochastic random fields for modeling and
characterizing spatial phenomena such as Earth surface features and geological structures.
Commonly used methods, such as the sequential Gaussian simulation (Gómez-Hernández and
Journel, 1993) and turning bands algorithms (Matheron, 1973), are based on kriging ( e.g.,
Graeler et al., 2016; Li and Heap, 2014; Tadić et al., 2017; 2015). This family of approaches
implies spatial relations using exclusively pairs of points and expresses these relations using



covariance functions. In the last two decades, multiple point statistics (MPS) emerged as a
method for representing more complex structures using high-order nonparametric statistics
(Guardiano and Srivastava, 1993). To do so, MPS algorithms rely on training images, which
are images with similar characteristics to the modeled area. Over the last decade, MPS has been
used for stochastic simulation of random fields in a variety of domains such as geological
modeling (e.g., Barfod et al., 2018; Strebelle et al., 2002), remote sensing data processing (e.g.,
Gravey et al., 2019; Yin et al., 2017), stochastic weather generation (e.g., Oriani et al., 2017;
Wojcik et al., 2009), geomorphological classification (e.g., Vannametee et al., 2014) and
climate model downscaling (a domain that has typically been the realm of kriging-based
methods ( e.g., Bancheri et al., 2018; Jha et al., 2015; Latombe et al., 2018)).
In the world of MPS simulations, one can distinguish two types of approaches. The first
category is the patch-based methods, where complete patches of the training image are imported
into the simulation. This category includes methods such as SIMPAT (Arpat and Caers, 2007)
and DISPAT (Honarkhah and Caers, 2010), which are based on building databases of patterns,
and image quilting (Mahmud et al., 2014), which uses an overlap area to identify patch
candidates, which are subsequently assembled using an optimal cut. CCSIM (Tahmasebi et al.,
2012) uses cross-correlation to rapidly identify optimal candidates. More recently, Li (2016)
proposed a solution that uses graph-cuts to find an optimal cut between patches, which has the
advantage of operating easily and efficiently independently of the dimensionality of the
problem. Tahmasebi (2017) propose as a solution that is based on "warping" in which the new
patch is distorted to match the previously simulated areas. For a multivariate simulation with
an informed variable, Hoffimann (2017) presented an approach for selecting a good candidate
based on the mismatch of the primary variable, and on the mismatch rank of the candidate
patches for auxiliary variables. Although patch-based approaches are recognized to be fast, they
typically suffer from a lack of variability due to the pasting of large areas of the training image,
which is a phenomenon that is called verbatim copy. Furthermore, patch-based approaches are
typically difficult to use in the presence of dense conditioning data.
The second category of MPS simulation algorithms consists of pixel-based algorithms, which
import a single pixel at the time instead of full patches. These methods are typically slower than
patch-based methods. However, they do not require a procedure for the fusion of patches, such
as an optimal cut, and they allow more flexibility in handling conditioning data. Furthermore,
in contrast to patch-based methods, pixel-based approaches rarely produce artifacts when
dealing with complex structures. The first pixel-based MPS simulation algorithm was
ENESIM, which was proposed by Guardiano and Srivastava, 1993, where for a given
categorical neighborhood – usually small – all possible matches in the training image are
searched. The conditional distribution of the pixel to be simulated is estimated based on all
matches, from which a value is sampled. This approach could originally handle only a few
neighbors and a relatively small training image; otherwise, the computational cost would
become prohibitive and the number of samples insufficient for estimating the conditional
distribution. Inspired by research in computer graphics, where similar techniques are developed
for texture synthesis (Mariethoz and Lefebvre, 2014), an important advance was the
development of SNESIM (Strebelle, 2002), which proposes storing in advance all possible
conditional distributions in a tree structure and using a multigrid simulation path to handle large


structures. With IMPALA, Straubhaar (2011) proposed reducing the memory cost by storing
information in lists rather than in trees. Another approach is direct sampling (DS) (Mariethoz
et al., 2010), where the estimation and the sampling of the conditional probability distribution
are bypassed by sampling directly in the training image, which incurs a very low memory cost.
DS enabled the first use of pixel-based simulations with continuous variables. DS can use any
distance formulation between two patterns; hence, it is well suited for handling various types
of variables and multivariate simulations.
In addition to its advantages, DS has several shortcomings: DS requires a threshold – which is
specified by the user – that enables the algorithm to differentiate good candidate pixels in the
training image from bad ones based on a predefined distance function. This threshold can be
highly sensitive and difficult to determine and often dramatically affects the computation time.
This results in unpredictable computation times, as demonstrated by Meerschman (2013). DS
is based on the strategy of randomly searching the training image until a good candidate is
identified (Shannon, 1948). This strategy is an advantage of DS; however, it can also be seen
as a weakness in the context of modern computer architectures. Indeed, random memory access
and high conditionality can cause 1) suboptimal use of the instruction pipeline, 2) poor memory
prefetch, 3) substantial reduction of the useful memory bandwidth and 4) impossibility of using
vectorization (John Paul Shen, 2018). While the first two problems can be addressed with
modern compilers and pseudorandom sequences, the last two are inherent to the current
memory and CPU construction.
This paper presents a new and flexible pixel-based simulation approach, namely, Quantile
Sampling (QS), which makes efficient use of modern hardware. Our method takes advantage
of the possibility of decomposing the standard distance metrics that are used in MPS ($L^0, L^2$) as
sums of cross-correlations. As a result, we can use fast Fourier transforms (FFTs) to quickly
compute mismatch maps. To rapidly select candidate patterns in the mismatch maps, we use an
optimized partial sorting algorithm. A free, open-source and flexible implementation of QS is
available, which is interfaced with most common programming languages (C/C++, MATLAB,
R, and Python 3).
The remainder of this paper is structured as follows: Section 2 presents the proposed algorithm
with an introduction to the general method of sequential simulation, the mismatch measurement
using FFTs and the sampling approach of using partial sorting followed by methodological and
implementation optimizations. Section 3 evaluates the approach in terms of quantitative and
qualitative metrics via simulations and conducts benchmark tests against DS, which is the only
other available approach that can handle continuous pixel-based simulations. Section 4
discusses the strengths and weaknesses of QS and provides guidelines. Finally, guidelines and
the conclusions of this work are presented in Section 5.





**2. Methodology and Implementation**
**2.1. Pixel-based sequential simulation**
We recall the main structure of pixel-based MPS simulation algorithms (Mariethoz and Caers,
2014, p.156), which is summarized and adapted for QS in Pseudocode 1. The key difference
between existing approaches is in lines 3 and 4 of Pseudocode 1, when candidate patterns are
selected. This task is the most time-consuming in many MPS algorithms and we focus only on
computing it in a way that reduces its cost and minimizes the parameterization.

| |
|---|
| 124   Pseudocode 1: QS Algorithm |
| 125 |
| 126   Inputs: |
| 127   $T$ the training images |
| 128   $S$ the simulation grid, including the conditioning data |
| 129   $P$ the simulation path |
| 130   The choice of pattern metric |
| 131 |
| 132   1. **For** each unsimulated pixel $x$ following the path $P$: |
| 133   2.     Find the neighborhood $N(x)$ in $S$ that contains all previously simulated or conditioning |
| 134         nodes in a specified radius |
| 135   3.     Compute the mismatch map between $T$ and $N(x)$: Section 2.3 |
| 136   4.     Select a good candidate using quantile sorting over the mismatch map: Section 2.4 |
| 137   5.     Assign the value of the selected candidate to $x$ in $S$ |
| 138   **6. End** |


**2.2. Decomposition of common mismatch metrics as sums of products**
Distance-based MPS approaches are based on pattern matching (Mariethoz and Lefebvre,
2014). Here, we rely on the observation that many common matching metrics can be expressed
as weighted sums of the pixelwise mismatch $\varepsilon$. This section explores the pixelwise errors for a
single variable and for multiple variables. For a single variable, the mismatch metric $\varepsilon$ between
two pixels is the distance between two scalars or two classes. In the case of many variables, it
is a distance between two vectors that are composed by scalars, by classes, or by a combination
of the two. Here, we focus on distance metrics that can be expressed in the following form:

*Equation 1*



$$\varepsilon(a,b) \propto \sum_j f_j(a).g_j(b)$$
where $a$ and $b$ represent the values of two univariate pixels and $f_j$ and $g_j$ are functions that
depend on the chosen metric. Here, we use the proportion symbol because we are interested in
relative metrics rather than absolute metrics, namely, the objective is to rank the candidate
patterns. We show below that many of the common metrics or distances that are used in MPS
can be expressed as Equation 1.
For the simulation of continuous variables, the most commonly used mismatch metric is the $L^2$-
norm, which can be expressed as follows:
*Equation 2*
$$\varepsilon_{L^2}(a,b) = (a-b)^2 = a^2 - 2ab + b^2$$
Using Equation 1, this $L^2$-norm can be decomposed into the following series of functions $f_j$ and
$g_j$:

| 161 | $f_0: x \to x^2$ | 164 | $g_0: x \to 1$ |
| 162 | $f_1: x \to -2x$ | 165 | $g_1: x \to x$ |
| 163 | $f_2: x \to 1$ | 166 | $g_2: x \to x^2$ |






A similar decomposition is possible for the $L^0$ -norm (also called Hamming distance), which is
commonly used for the simulation of categorical variables. This measure of dissimilarity counts
the number of nonzero values in a vector (Hamming, 1950)
*Equation 3*
$$\varepsilon_{L^0}(a,b) = (a-b)^0 = \sum_{j \in \mathcal{C}} 1 - \left(\delta_{a,j}.\delta_{b,j}\right) \propto \sum_{j \in \mathcal{C}} \delta_{a,j}.\delta_{b,j}$$

where $\delta_{x,y}$ is the Kronecker delta between $x$ and $y$, which is equal to 1 if $x$ equals $y$ and 0
otherwise, and $\mathcal{C}$ is the set of all possible categories of a specified variable.
Using Equation 1, this $L^0$ distance can be decomposed (Arpat and Caers, 2007) into the
following series of functions $f_j$ and $g_j$ :
$f_j: x \rightarrow -\delta_{xj}$
$g_j: x \rightarrow \delta_{xj}$
with a new pair of $f_j$ and $g_j$ for each class $j$ of $\mathcal{C}$.
For multivariate pixels, such as a combination of categorical and continuous values, the
mismatch $\varepsilon$ can be expressed as a sum of univariate pixelwise mismatches.
*Equation 4*
$$\varepsilon(\boldsymbol{a},\boldsymbol{b}) \propto \sum_i \sum_j f_j(a_i).g_j(b_i)$$

where $\boldsymbol{a}$ and $\boldsymbol{b}$ are the compared vectors and $a_i$ and $b_i$ are the individual components of $\boldsymbol{a}$ and
$\boldsymbol{b}$.

## 2.3. Computation of a mismatch map for an entire pattern
The approach that is proposed in this work is based on computing a mismatch map in the TI for
each simulated pixel. The mismatch map is a grid that represents the pattern-wise mismatch for
each location of the training image and enables the fast identification of a good candidate, as
shown by the red circle in Figure 1.



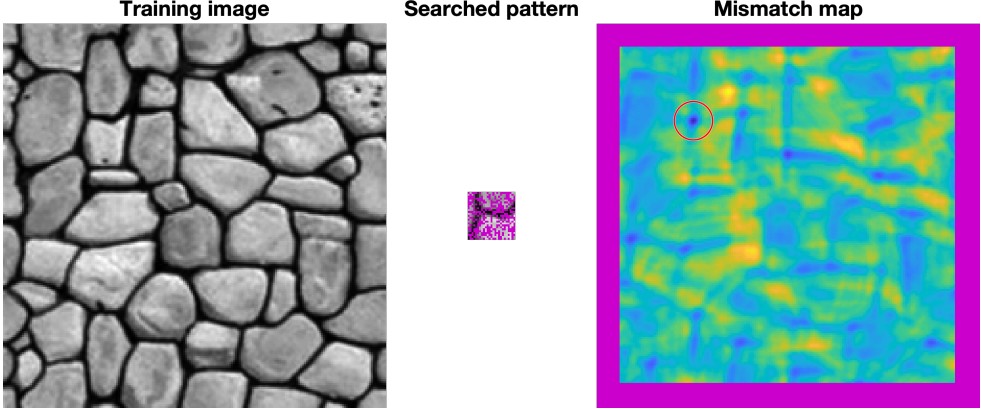

**Training image**   **Searched pattern**   **Mismatch map**


*Figure 1 Example of a mismatch map for an incomplete pattern. Blue represents good matches*
*and yellow bad matches. The red circle highlights the minimum of the mismatch map, which*
*corresponds to the location of the best candidate.*

If we consider the neighborhood $N(s)$ around the simulated position $s$, then we can express a
weighted dissimilarity between $N(s)$ and a location in the TI $N(t)$:

*Equation 5*

$$\mathrm{E}\big(N(t), N(s)\big) = \sum_{l \mid N_l(t) \ and \ N_l(s) \ exist} \omega_l \varepsilon\big(N_l(t), N_l(s)\big)$$

where $N_l(p)$ is the ensemble of neighbors of $p$ ($p$ can represent either $s$ or $t$), $l$ is the lag vector
that defines the relative position of each value within $N$, and $\omega_l$ is a weight for each pixelwise
error according to the lag vector $l$. By extension, $\omega$ is the matrix of all weights, which we call
the weighting kernel or, simply, the kernel. E represents the mismatch between patterns that are
centered on $s$ and $t \in T$, where $T$ is the training image.
Some lags may not correspond to a value, for example, due to edge effects in the considered
images or because the patterns are incomplete. Missing patterns are inevitable during the course
of a simulation using a sequential path. Furthermore, in many instances, there can be missing
areas in the training image. This is addressed by creating an indicator variable to be used as a
mask, which equals 1 at informed pixels and 0 everywhere else:

*Equation 6*

$$\mathbb{1}_l(p) = \begin{cases} 1 \ if \ N_l(p) \ is \ informed \\ 0 \ otherwise \end{cases}$$

Let us first consider the case in which for a specified position, either all or no variables are
informed. Expressing the presence of data as a mask enables the gaps to be ignored because the
corresponding errors are multiplied by zero.
Then, Equation 5 can be expressed as follows:

*Equation 7*





$$E\big(N(t), N(s)\big) = \sum_l \omega_l . \mathbb{1}_l(t) . \mathbb{1}_l(s) . \varepsilon\big(N_l(t), N_l(s)\big)$$

. Combining Equation 4 and Equation 7, we get:
*Equation 8*
$$E\big(N(t), N(s)\big) \propto \sum_l \omega_l . \mathbb{1}_l(t) . \mathbb{1}_l(s) \sum_j \sum_i f_j(N_l(t)_i) . g_j(N_l(s)_i)$$

$$= \sum_l \sum_j \sum_i \omega_l . \mathbb{1}_l(t) . \mathbb{1}_l(s) . f_j(N_l(t)_i) . g_j(N_l(s)_i)$$

$$= \sum_i \sum_j \sum_l \omega_l . \big(\mathbb{1}_l(t) . f_j(N_l(t)_i)\big) . \big(\mathbb{1}_l(s) . g_j(N_l(s)_i)\big)$$

.

After rewriting and reordering, Equation 8 can be expressed as a sum of cross-correlations that
encapsulate spatial dependencies:
*Equation 9*
$$E\big(N(t), N(s)\big) \propto \sum_i \sum_j \big(\mathbb{1}(t) \circ f_j(N(t)_i)\big) \star \big(\omega \circ \mathbb{1}(s) \circ g_j(N(s)_i)\big)$$

, where $\omega$ and $\mathbb{1}(.)$ represent the matrices that are formed by $\omega_l$ and $\mathbb{1}_l(.)$ for all possible vectors
$l$, $\star$ denotes the cross-correlation operator, and $\circ$ is the element-wise product (or Hadamard-
product).
Finally, by applying cross-correlations for all positions $t \in T$, we obtain a mismatch map,
which is expressed as:
*Equation 10*
$$E\big(T, N(s)\big) \propto \sum_i \sum_j \big(\mathbb{1}(T) \circ f_j(T_i)\big) \star \big(\omega \circ \mathbb{1}(s) \circ g_j(N(s)_i)\big)$$

. The term $\mathbb{1}(T)$ allows the consideration of the possibility of missing data in the training image
$T$.
Let us consider the general case in which only some variables are informed and the weighting
can vary for each variable. Equation 10 can be extended for this case by defining separate masks
and weights $\omega_i$ for each variable:
*Equation 11*
$$E\big(T, N(s)\big) \propto \sum_i \sum_j \big(\mathbb{1}(T_i) \circ f_j(T_i)\big) \star \big(\omega_i \circ \mathbb{1}(s_i) \circ g_j(N(s)_i)\big)$$

. Equation 11 can be expressed using the convolution theorem:
*Equation 12*



$$\mathrm{E}\big(T, N(s)\big) \propto \sum_i \sum_j \mathcal{F}^{-1}\left\{\overline{\mathcal{F}\{\mathbb{1}(T_i) \circ f_j(T_i)\}} \circ \mathcal{F}\{\omega_i \circ \mathbb{1}(s_i) \circ g_j(N(s)_i)\}\right\}$$

, where $\mathcal{F}$ represents the Fourier transform, $\mathcal{F}^{-1}$ the inverse transform, and $\bar{x}$ the conjugate of
$x$.
By linearity of the Fourier transform, the summation can be performed in Fourier space, thereby
reducing the number of transformations:

*Equation 13*

$$\mathrm{E}\big(T, N(s)\big) \propto \mathcal{F}^{-1}\left\{\sum_i \sum_j \overline{\mathcal{F}\{\mathbb{1}(T_i) \circ f_j(T_i)\}} \circ \mathcal{F}\{\omega_i \circ \mathbb{1}(s_i) \circ g_j(N(s)_i)\}\right\}$$

. Equation 13 is appropriate for modern computers, which are well-suited for computing FFTs
(Cooley et al., 1965; Gauss, 1799). Currently, FFTs are well implemented in highly optimized
libraries (Rodríguez, 2002). Equation 13 is the expression that is used in our QS implementation
because it reduces the number of Fourier transforms, which are the most computationally
expensive operations of the algorithm. One issue with the use of FFTs is that the image $T$ is
typically assumed to be periodic. However, in most practical applications, it is not periodic.
This can be simply addressed by cropping the edges of $E\big(T, N(s)\big)$ or by adding a padding
around $T$.
The computation of the mismatch map (Equation 13) is deterministic; as a result, it incurs a
constant computational cost that is independent of the pixel values. Additionally, Equation 13
is expressed without any constraints on the dimensionality. Therefore, it is possible to use the
$n$-dimensional FFTs that are provided in the above libraries to perform $n$-dimensional
simulations without changing the implementation.
## 2.4. Selection of candidates based on a quantile
The second contribution of this work is the $k$-sampling strategy for selecting a simulated value
among candidates. The main idea is to use the previously calculated mismatch map to select a
set of potential candidates that are defined by the $k$ smallest (i.e. a quantile) values of E. Once
this set has been selected, we randomly draw a sample from this pool of candidates. This differs
from strategies that rely on a fixed threshold, which can be cumbersome to determine. This
strategy is highly similar to the ε-replicate strategy that is used in image quilting (Mahmud et
al., 2014) in that we reuse and extend to satisfy the specific requirements of QS. It has the main
advantage of rescaling the acceptance criterion according to the difficulty; i.e. the algorithm is
more tolerant of rare patterns while requiring very close matches for common patterns.
In detail, the candidate selection procedure is as follows: All possible candidates are ranked
according to their mismatch and one candidate is randomly sampled among the $k$ best. This
number $k$ can be seen as a quantile over the training dataset. However, parameter $k$ has the
advantage of being an easy representation for users, who can associate $k = 1$ with the best
candidate, $k = 2$ with the two best candidates, etc. For fine-tuning parameter $k$, the sampling



strategy can be extended to noninteger values of $k$ by sampling the candidates with probabilities
that are not uniform. For example, if the user sets $k = 1.5$, the best candidate has a probability
of 2/3 of being sampled and the second best a probability of 1/3. For $k = 3.2$, (Figure 2) each
of the 3 best candidates are sampled with an equal probability of 0.3125 and the 4[th] best with a
probability of 0.0625. This feature is especially useful for tuning $k$ between 1 and 2 and for
avoiding a value of $k = 1$, which can result in the phenomenon of verbatim copy.

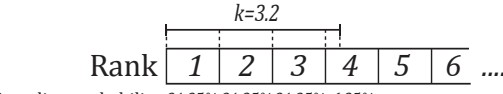


*Figure 2 Illustration of the k-sampling strategy*
An alternative sampling strategy for reducing the simulation time is presented in Appendix A.3.
However, this strategy can result in a reduction in the simulation quality.

### 2.5. Simplifications in the case of a fully informed training image

In many applications, spatially exhaustive TIs are available. In such cases, the equations above
can be simplified by dropping constant terms from Equation 1, thereby resulting in a simplified
form for Equation 13. Here, we take advantage of the ranking to know that a constant term will
not affect the result.
As in Tahmasebi (2012), in the $L^2$-norm, we drop the squared value of the searched pattern,
namely, $b^2$, from Equation 2. Hence, we can express Equation 4 as follows:

*Equation 14*

$$\varepsilon(\boldsymbol{a}, \boldsymbol{b}) = \sum_i a_i^2 - 2 \sum_i a_i . b_i$$

The term $a^2$, which represents the squared value of the candidate pattern in the TI, differs
among training image locations and, therefore, cannot be removed. Indeed, the assumption that
$\sum a^2$ is constant is only valid under a strict stationarity hypothesis on the scale of the search
pattern. While this hypothesis might be satisfied in some cases (as in Tahmasebi et al., 2012),
we do not believe it is generally valid. Via the same approach, Equation 3 can be simplified by
removing the constant terms; then, we obtain the following for the $L^0$-norm:

*Equation 15*

$$\varepsilon(\boldsymbol{a}, \boldsymbol{b}) = -\sum_{j \in \mathcal{C}} \sum_i \delta_{a_i, j} . \delta_{b_i, j}$$

.



### 2.6. Efficient Implementation


An efficient implementation of QS was achieved by 1) performing precomputations, 2)
implementing an optimal partial sorting algorithm for selecting candidates and 3) optimal
coding and compilation. These are described below.
According to Equation 13, $\overline{\mathcal{F}\{\mathbb{1}(T_i) \circ f_j(T_i)\}}$ is independent of the searched pattern $N(s)$.
Therefore, it is possible to precompute it at the initialization stage for all $i$ and $j$. This
improvement typically reduces the computation time for an MPS simulation by a factor of at
least 2.
In the QS algorithm, a substantial part of the computation cost is incurred in identifying the $k$
best candidates in the mismatch map. In the case of noninteger $k$, the upper limit $\lceil k \rceil$ is used.
Identifying the best candidates requires sorting the values of the mismatch map and retaining
the candidates in the top $k$ ranks. For this, an efficient sorting algorithm is needed. The
operation of finding the $k$ best candidates can be implemented with a partial sort, in which only
the elements of interest are sorted, while the other elements remain unordered. This results in
two sets: $\mathfrak{S}_s$ with the $k$ smallest elements and $\mathfrak{S}_l$ with the largest elements. The partial sort
guarantees that $x \leq y \mid (x, y) \in \mathfrak{S}_s \times \mathfrak{S}_l$. More information about our implementation of this
algorithm is available in Appendix A.1. Here, we use a modified vectorized online heap-based
partial sort (Appendix A.1). With a complexity of $O(n. \ln(k))$, it is especially suitable for small
values of $k$. Using the cache effect, the current implementation yields results that are close to
the search of the best value (the smallest value of the array). The main limitation of standard
partial sort implementations is that in the case of equal values, either the first or the last element
is sampled. Here, we develop an implementation that can uniformly sample a position among
similar values with a single scan of the array. This is important because systematically selecting
the same position for the same pattern will reduce the conditional probability density function
to a unique sample, thereby biasing the simulation.
Due to the intensive memory access by repeatedly scanning large training images, interpreted
programming languages, such as MATLAB and Python, are inefficient for a QS
implementation and, in particular, for a parallelized implementation. We provide a NUMA-
aware and flexible C/C++/OpenMP implementation of QS that is highly optimized. Following
the denomination of Mariethoz (2010), we use a path-level parallelization with a waiting
strategy, which offers a good trade-off between performance and memory requirements. In
addition, two node-level parallelization strategies are available: if many training images are
used, a first parallelization is performed over the exploration of the training images; then, each
FFT of the algorithm is parallelized using natively parallel FFT libraries.
The FFTw library (Frigo and Johnson, 2018) provides a flexible and performant architecture-
independent framework for computing $n$-dimensional Fourier transformations. However, an
additional speed gain of approximately 20% was measured by using the Intel MKL library (Intel
Corporation, 2019) on compatible architectures. We also have a GPU implementation that uses
clFFT for compatibility. Many Fourier transforms are sparse and, therefore, can easily be

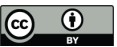



accelerated in *n*-dimensional cases with "partial FFT" since Fourier transforms of only zeros
result in zeros.

## 3. Results

### 3.1. Simulation examples

This section presents illustrative examples for continuous and categorical case studies in 2D
and in 3D. Additional tests are reported in Appendix 0. The parameters that are used for the
simulations of Figure 3 are reported in Table 1.
The results show that simulations results are consistent with what is typically observed with
state-of-the-art MPS algorithms. While simulations can accurately reproduce TI properties for
relatively standard examples with repetitive structures (e.g., MV, Strebelle, and Folds), training
images with long-range features (typically larger than the size of the TI) are more difficult to
reproduce, such as in the Berea example. For multivariate simulations, the reproduction of the
joint distribution is satisfactory, as observed in the scatter plots (Figure 3).




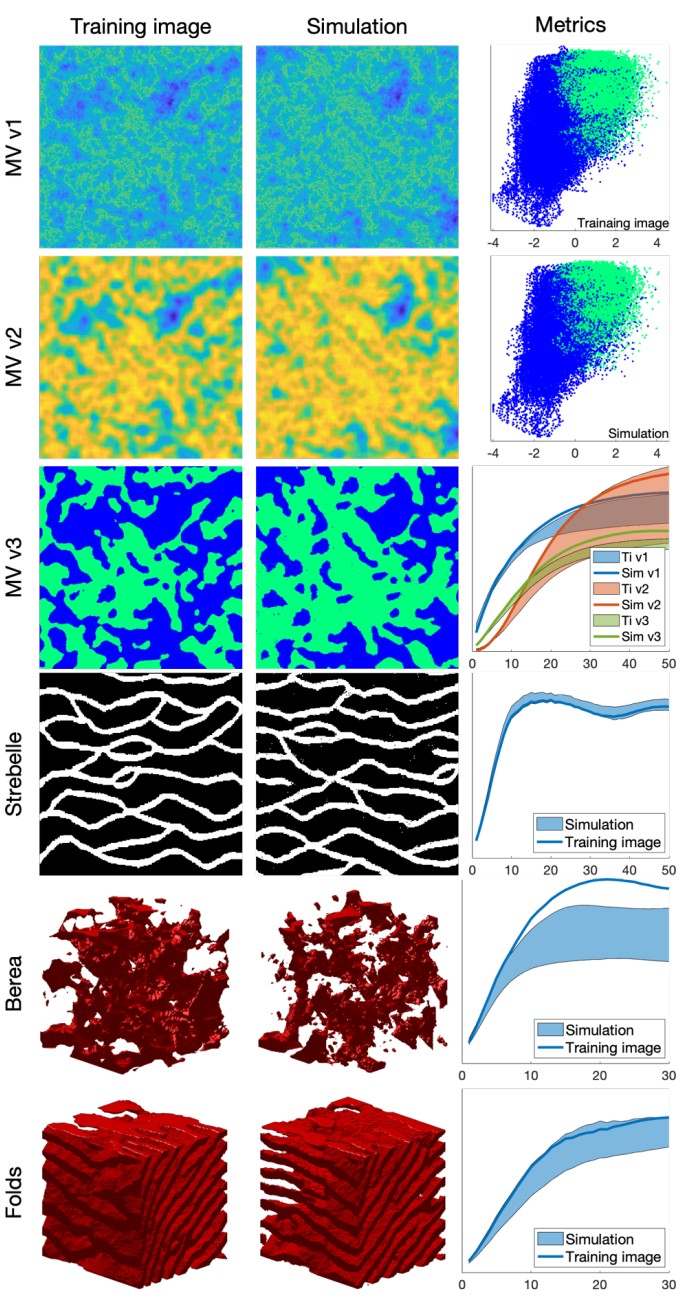


*Figure 3 Examples of unconditional continuous and categorical simulations in 2D and 3D and*
*their variograms. The first column shows the training images that were used, the second column*
*one realization, and the third column quantitative quality metrics. MVs v1, v2 and v3 represent*
*a multivariate training image (and the corresponding simulation) using 3 variables. The first*
*two metrics are scatter plots of MV v1 vs. MV v2 of the training image and the simulation,*
*respectively. The third metric represents the reproduction of the variogram for each of MVs v1,*
*v2 and v3.*




|  | MVs v1, v2, v3 | Strebelle | Berea | Folds |
|---|---|---|---|---|
| Source | (Mariethoz and Caers, 2014) | (Strebelle, 2002) | Doi:10.6084/m9.figshare.1153794 | (Mariethoz and Caers, 2014) |
| Size of the training image (px) | $490 \times 490$ | $250 \times 250$ | $100 \times 100 \times 100$ | $180 \times 150 \times 120$ |
| Size of the simulation (px) | $490 \times 490$ | $250 \times 250$ | $100 \times 100 \times 100$ | $180 \times 150 \times 120$ |
| Computation time (s) | 1456 | 54 | 1665 | 76270 |
| $k$ | 1.2 | | | |
| $N$ | 80 | | 125 | |

*Table 1 Parameters that were used for the simulations in Figure 3. Times are specified for*
*simulations without parallelization.*

## 3.2. Comparison with direct sampling simulations

QS simulations are benchmarked against DS using the "Stone" training image (Figure 4). The
settings that are used for DS are based on optimal parameters that were obtained via the
approach of Baninajar et al. (2019), which uses stochastic optimization to find optimal
parameters. In DS, we use a fraction of scanned TI of $f = 1$ to explore the entire training image
via the same approach as in QS and we use the $L^2$-norm as in QS. To avoid the occurrence of
verbatim copy, we include 0.1% conditioning data, which are randomly sampled from a rotated
version of the training image. The number of neighbors $N$ is set to 20 for both DS and QS and
the acceptance threshold of DS is set to 0.001.
The comparison is based on qualitative (Figure 5) and quantitative (Figure 6) metrics, which
include directional and omnidirectional variograms, along with the connectivity function and
the Euler characteristic (Renard and Allard, 2013). The results demonstrate that the simulations
are of a quality that is comparable to DS. With extreme settings (highest pattern reproduction
regardless of the computation time), both algorithms perform similarly, which is reasonable
since both are based on sequential simulation and both directly import data from the training
image.
With QS, kernel weighting enables the adaption of the parametrization to improve the results,
as shown in Figure 5. In this paper, we use an exponential kernel:

*Equation 16*

$$\omega_l = e^{-\alpha \|l\|_2}$$



where $\alpha$ is a kernel parameter. The validation metrics of Figure 6 show that both QS and DS
tend to slightly underestimate the variance and the connectivity. Figure 6 shows that an optimal
kernel improves the results for all metrics, with all training image metrics in the 5-95%
realization interval, except for the Euler characteristic.

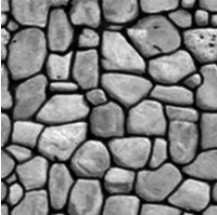


*Figure 4 Training image that was used for benchmarking and sensitivity analysis.*

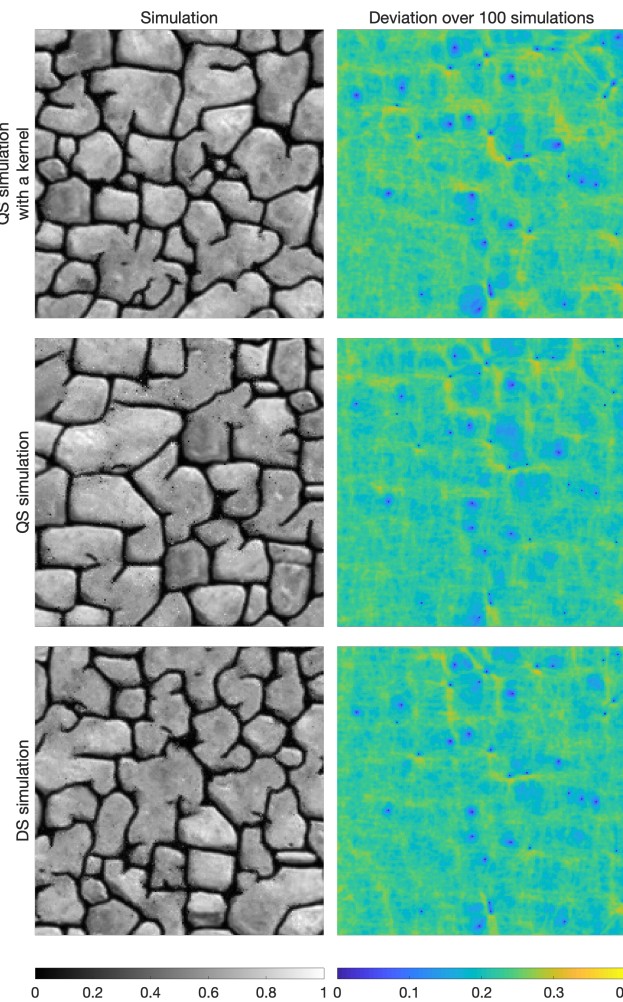


*Figure 5 Examples of conditional simulations and their standard deviation over 100*
*realizations that are used in the benchmark between QS and DS.*





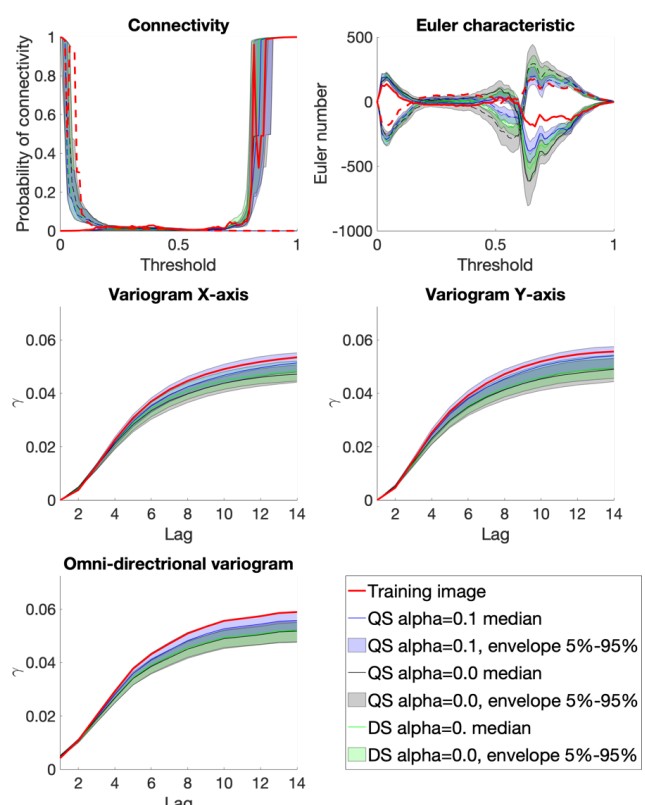

*Figure 6 Benchmark between QS and DS over 5 metrics.*

## 3.3. Parameter sensitivity analysis

In this section, we perform a sensitivity analysis on the parameters of QS using the training image in Figure 4. Only essential results are reported in this section (Figure 7 and Figure 8); more exhaustive test results are available in Appendix 0 (Figure A 4 and Figure A 5). The two main parameters of QS are the number of neighbors $N$ and the number of used candidates $k$.

Figure 7 (and Appendix 0 Figure A 4) shows that large $N$ values and small $k$ values improve the simulation performance; however, tend to induce verbatim copy in the simulation. Small values of $N$ result in noise with good reproduction of the histogram.



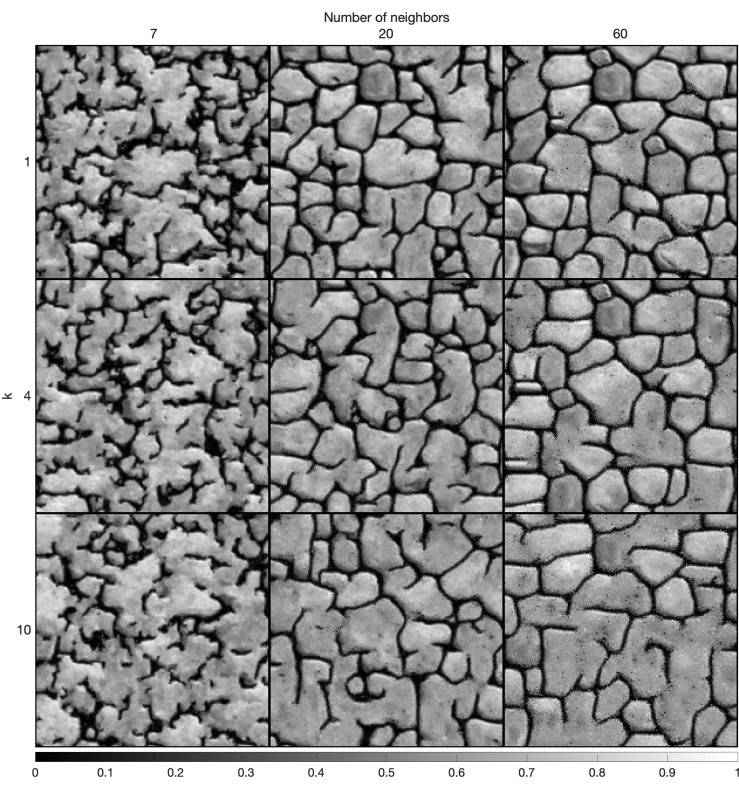


*Figure 7 Sensitivity analysis on one simulation for the two main parameters of QS using a*
*uniform kernel.*
$\omega$ can be a very powerful tool, typically using the assumption that the closest pixels are more
informative than remote pixels. The results of study of the effect of the kernel value $\alpha$ are
explored in Figure 8 and Figure A 5, which shows that $\alpha$ provides a unique tool for improving
the quality of the simulation.

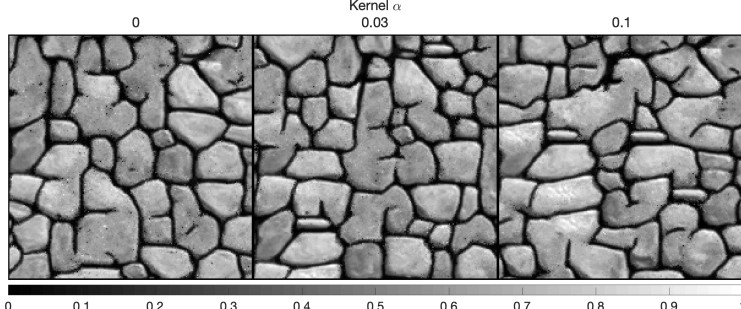


*Figure 8 Sensitivity analysis on the kernel parameter $\alpha$, with fixed parameters k=1.5 and N=40.*



### 3.4. Computational efficiency and scalability

In this section, we investigate the scalability of QS with respect to the size of the simulation grid, the size of the training image grid, the number of variables, incomplete training images, and hardware. According to the test results, the code will continue to scale with new-generation hardware.

As explained in Section 2.3 and 2.4, the amounts of time that are consumed by the two main operations of QS (finding candidates and sorting them) are independent of the pixel values. Therefore, the training image that is used is not relevant (here, we use simulations that were performed with the TI of Figure 4 and its classified version for categorical cases). Furthermore, the computation time is independent of the parametrization ($k$ and $N$). However, the performance is affected by the type of mismatch function that is used; here, we consider both continuous (Equation 2 and Equation 14) and categorical cases (Equation 3 and Equation 15).

We also test our implementation on different types of hardware, as summarized in Table 2. We expect Machine (2) to be faster than Machine (1) for medium-sized problems due to the high memory bandwidth requirement of QS. Machine (3) should also be faster than Machine (1) because it takes advantage of a longer vector computation (512-bit VS. 256-bit instruction set).

| Name of the machine | Machine (1) | Machine (2) | Machine (3) |
|---|---|---|---|
| CPU | -2x Intel(R) Xeon(R) CPU E5-2680 v2 @ 2.80 GHz | -Xeon Phi, Intel(R) Xeon Phi (TM) CPU 7210 @ 1.30 GHz | -2x Intel(R) Xeon(R) Gold 6128 Processor @ 3.40 GHz |
| Memory type | - DDR3 | - MCDRAM / DDR4 | - DDR4 |
| OS, compiler and compilation flags | Linux, Intel C/C++ compiler 2018 with -xhost | | |

*Table 2 Hardware that was used in the experiments*

Figure 9 plots the execution times on the 3 tested machines for continuous and categorical cases and with training images of various sizes. Since QS has a predictable execution time, the influence of the parameters on the computation time is predictable: linear with respect to the number of variables (Figure 9a, Figure 9b), linear with respect to the size of the simulation grid and following a power function of the size of the training image (Figure 9c). Therefore, via a few tests on a set of simulations, one can predict the computation time for any other setting.

Figure 9d shows the scalability of the algorithm when using the path-level parallelization. The algorithm scales well until all physical cores are being used. Machine (3) has a different scaling factor (slope). This suboptimal scaling is attributed to the limited memory bandwidth. Our implementation of QS scales well with an increasing number of threads (Figure 9d), with an efficiency above 80% using all possible threads. The path-level parallelization strategy that was used involves a bottleneck for large number of threads due to the need to wait for neighborhood conflicts to be resolved (Mariethoz 2010). This effect typically appears for large values of $N$ or



intense parallelization (>50 threads) on small grids. It is assumed that small grids do not require
intense parallelization; hence, this problem is irrelevant in most applications.

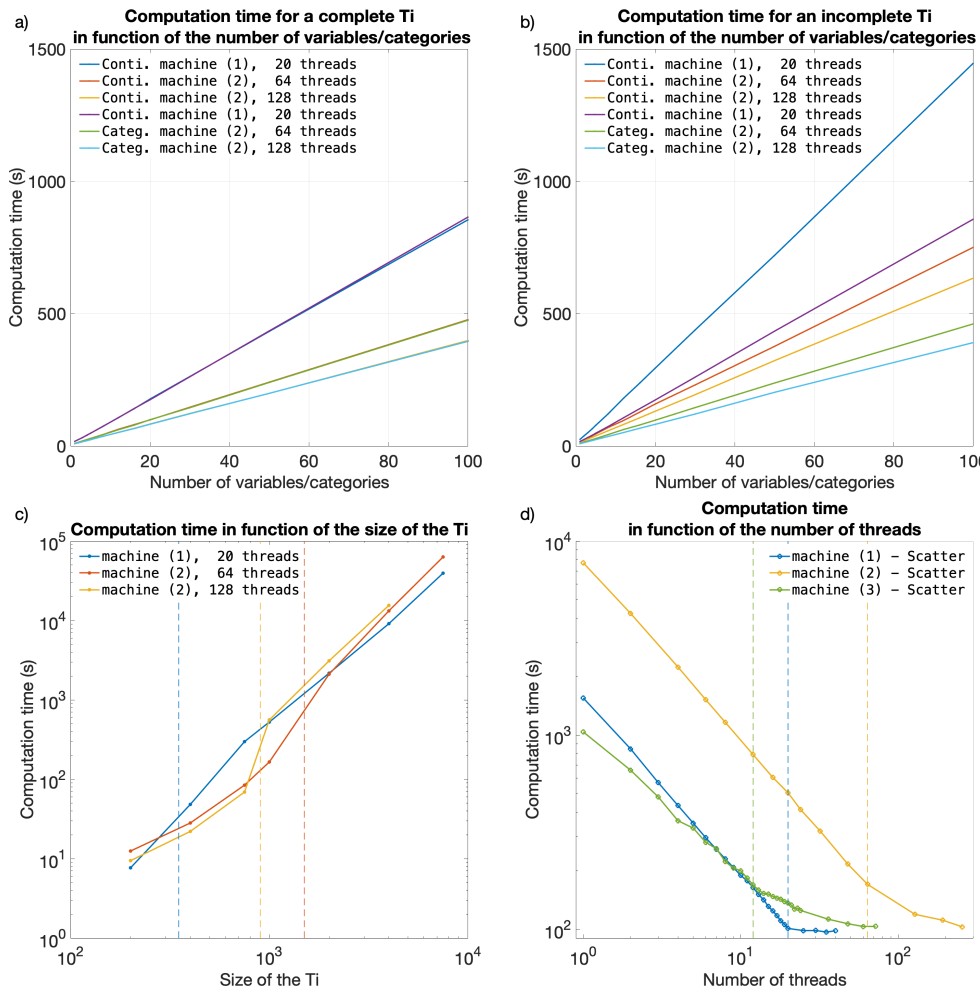


*Figure 9 Efficiency of QS with respect to all key parameters. a) and b) are the evolutions of the*
*computation time for complete and incomplete training images, respectively, with continuous*
*and categorical variables. c) shows the evolution of the computation time as the size of the*
*training image is varied; the dashed lines indicate that the training image no longer fits in the*
*CPU cache. d) shows the evolution of the computation time as the number of threads is*
*increased. The dashed lines indicate that all physical cores are used.*

## 4. Discussion

The parameterization of the algorithm (and therefore simulation quality) has almost no impact
on the computational cost, which is an advantage. Indeed, many MPS algorithms impose trade-





offs between the computation time and the parameters that control the simulation quality,
thereby imposing difficult choices for the users. QS is comparatively simpler to set up in this
regard. In practice, a satisfactory parameterization strategy is often to start with a small $k$ value
(say 1.2) and a large $N$ value ($> 50$) and then gradually change these values to increase the
variability if necessary (Figure 6 and Figure A 4).
QS is adapted for simulating continuous variables using the $L^2$-norm. However, a limitation is
that the $L^1$-norm does not have a decomposition that satisfies Equation 1 and, therefore, cannot
be used with QS. Another limitation is that for categorical variables, each class requires a
separate FFT, which incurs an additional computational cost. This renders QS less
computationally efficient for categorical variables (if there are more than 2 categories) than for
continuous variables. For accelerated simulation of categorical variables, a possible alternative
to reduce the number of required operations is presented in Appendix A.2. The strategy is to
use encoded variables, which are decoded in the mismatch map. While this alternative yields
significant computational gains, it does not allow the use of a kernel weighting and is prone to
numerical precision issues.
The computational efficiency of QS is generally great compared to other pixel-based
algorithms: for example, in our tests it performed faster than DS. QS requires more memory
than DS, especially for applications with categorical variables with many classes and with a
path-level parallelization. However, the memory requirement is much lower compared to MPS
algorithms that are based on a pattern database, such as SNESIM.
There may be cases in which it is slower than DS, in particular when using a large training
image that is highly repetitive. In such cases, using DS can be advantageous as it must scan
only a very is small part of the training image. For scenarios of this type, it is possible to adapt
QS such that only a small subset of the training image is used; this approach is described in
Appendix A3.
QS can be extended to handle the rotation and scaling of patterns by applying a constant rotation
or affinity transformation to the searched patterns (Strebelle, 2002). However, the use rotation-
invariant distances and affinity-invariant distances (as in Mariethoz and Kelly, 2011), while
possible in theory, would substantially increase the computation time. Mean-invariant distances
can be implemented by simply adapting the distance formulation in QS. All these advanced
features are outside the scope of this paper.
## 5. Conclusions
QS is an alternative approach for performing $n$-dimensional pixel-based simulations, which
uses an $L^2$-distance for continuous cases and an $L^0$-distance for categorical data. The framework
is highly flexible and allows other metrics to be used. The simple parameterization of QS
renders it easy to use for nonexpert users. Compared to other pixel-based approaches, QS has
the advantage of generating realizations in constant and predictable time for a specified training
image size. Using the quantile as a quality criterion naturally reduces the small-scale noise
compared to DS. In terms of parallelization, the QS code scales well and can adapt to new
architectures due to the use of external highly optimized libraries.





The QS framework provides a complete and explicit mismatch map, which can be used to
formulate problem-specific rules for sampling or even solutions that take the complete
conditional probability density function into account, for example, such as a narrowness
criterion for the conditional pdf of the simulated value (Gravey et al., 2019; Rasera et al., 2019),
or to use the mismatch map to infer the optimal parameters of the algorithm.

## 6. Code availability

The source code and documentation of the QS simulation algorithm are available as part of the
G2S package at: https://github.com/GAIA-UNIL/G2S under GPLv3 license. Or permanently
at https://doi.org/10.5281/zenodo.3546338
Platform: Linux / macOS / Windows 10 Language: C/C++
Interfacing functions in MATLAB, Python3, R
A package is available with our unbiased partial sort at:
https://github.com/mgravey/randomKmin-max

## 7. Author contribution

MG proposed the idea, implemented and optimized the QS approach and wrote the manuscript.
GM provided supervision, methodological insights and contributed to the writing of the
manuscript.

## 8. Appendices

### A.1.   Partial sorting with random sampling

Standard partial sorting algorithms resolve tie ranks deterministically, which does not accord
with the objective of stochastic simulation with QS, where variability is sought. Here, we
propose an online heap-based partial sort. It is realized with a single scan of the array of data
using a heap to store previously found values. This approach is especially suitable when we are
interested in a small fraction of the entire array.
Random positions of the $k$ best values are ensured by swapping similar values. If $k = 1$, the
saved value is switched with a smaller value each time it is encountered. If an equal value is
scanned, a counter $c$ is increased for this specific value and a probability of $1/c$ of switching to
the new position is applied. If $k > 1$, the same strategy is extended by carrying over the counter
$c$.
This partial sort outperforms random exploration of the mismatch map. However, it is difficult
to implement efficiently on GPUs. A solution is still possible for shared-memory GPUs by
performing the partial sort on the CPU. This is currently available in the proposed
implementation.

---

$k$: the number of values of interest
$D$: the input data array
$S$: the array with the $k$ smallest values (sorted)
$Sp$: the array with the positions that are associated with the values of $S$

1.  **for** each value $v$ of $D$
2.       **if** $v$ is smaller than the smallest value of $S$
3.            search in $S$ for the position $p$ at which to insert $v$ and insert it
4.            **if** $p = k$           // last position of the array
5.                reinitialize the counter $c$ to 0
6.                insert $v$ at the last position
7.            **else**
8.                increment $c$ by one
9.                swap the last position with another of the same value
10.              insert the value at the expected position $p$
11.            **end**
12.       **else if** $v$ is equal to the smallest value of $S$
13.            increment $c$ by one
14.            change the position of $v$ to one of the $n$ positions of equal value with a probability of
$n/(n + c)$
15.       **end**
16. **end**

---

## A.2.     Encoded categorical variables
To handle categorical variables, a standard approach is to consider each category as an
independent variable. This requires as many FFTs as classes. This solution renders it expensive
to use QS in cases with multiple categories.
An alternative approach is to encode the categories and to decode the mismatch from the cross-
correlation. It has the advantage of only requiring only a single cross-correlation for each
simulated pattern.
Here, we propose encoding the categories as powers of the number of neighbors, such that their
product is equal to one if the class matches. In all other cases, the value is smaller than one or
larger than the number of neighbors.
$$\varepsilon_{L^0}(a, b) = \psi\big((a - b)^0 \propto -(N + 1)^{-p(a)} \cdot (N + 1)^{-p(b)}\big)$$

where $N$ is the largest number of neighbors that can be considered and $p(c)$ is an arbitrary
function that maps index classes of $\mathcal{C}$, $c \in \mathcal{C}$.
In this scenario, in Equation 1 this encoded distance $L_e^0$ can be decomposed into the following
series of functions $f_j$ and $g_j$ :
$f_0 \colon x \to -(N + 1)^{p(x)}$


$g_0: x \to (N+1)^{-p(x)}$
and the decoding function is

$\psi(x) = \lfloor x \rfloor \bmod N$

Table A 1 describes this process for 3 classes, namely, $a, b,$ and $c,$ and a maximum of 9
neighbors. Then, the error can be easily decoded by removing decimals and dozens.

| Products | $g_0(a) = 1$ | $g_0(b) = 0.1$ | $g_0(c) = 0.01$ |
|---|---|---|---|
| $f_0(a) = 1$ | **1** | 0.1 | 0.01 |
| $f_0(b) = 10$ | 10 | **1** | 0.1 |
| $f_0(c) = 100$ | 100 | 10 | **1** |

*Table A 1 Example of encoding for 3 classes and 9 neighbors and their associated products*

Consider the following combination:
$f_0(a, \quad b, \quad a, \quad c, \quad c, \quad b, \quad a, \quad a, \quad b)$
$\times g_0(c, \quad b, \quad b, \quad a, \quad a, \quad b, \quad c, \quad a, \quad a)$
$-(0.01, \quad 1, \quad 0.1, 100, 100, \quad 1, 0.01, \quad 1, \quad 10) = -213.12$
The decoding $\lfloor -213.12 \rfloor \bmod 10 = -213 \bmod 10 = -3$ yields 3 matches (in green).
This encoding strategy provides the possibility of drastically reducing the number of FFT
computations. However, the decoding phase is not always implementable if a nonuniform
matrix $\omega$ is used. Finally, the test results show that the method suffers quickly from numerical
precision issues, especially with many classes.
### A.3.      Sampling strategy using training image splitting
The principle of considering a fixed number of candidates can be extended by instead of taking
the $k^{th}$ best candidate, sampling the best candidate in only a portion $\frac{1}{k}$, of the TI. For instance,
as an alternative to considering $k = 4$, this strategy searches for the best candidate in one fourth
of the image. This is more computationally efficient. However, if all the considered candidates
are contiguous (by splitting the TI in $k$ chunks), this approximation is only valid if the TI is
completely stationary and all $k$ equal subdivisions of the TI are statistically identical. In
practice, real-world continuous variables are often nonstationary. However, in categorical
cases, especially in binary ones, the number of pattern replicates is higher and this sampling
strategy could be interesting.
The results of applying this strategy are presented in Table A 2 and Figure A 1. The
experimental results demonstrate that the partial exploration approach that is provided by
splitting substantially accelerates the processing time. However, Figure A 1 shows that the
approach has clear limitations when dealing with training images with complex and



nonrepetitive patterns. The absence of local verbatim copy can explain the poor-quality
simulation results.

Using all chunks          Using one random chunk

Berea

Folds

Strebelle


*Figure A 1 Comparison of QS using the entire training image and using training image*
*splitting. In these examples, the training image is split into two images over each dimension.*
*The original training images are presented in Figure 2.*




| Training image | Using all chunks | Using one random chunk | Speedup |
|---|---|---|---|
| Berea | 11 052 s | 1 452 s | 7.61x |
| Folds | 35 211 s | 4 063 s | 8.66x |
| Strebelle | 7.95 s | 3.16 s | 2.51x |

*Table A 2 Computation times and speedups for the full and partial exploration approaches.*
*Times are specified for simulations with path level parallelization.*
## A.4.    Additional results

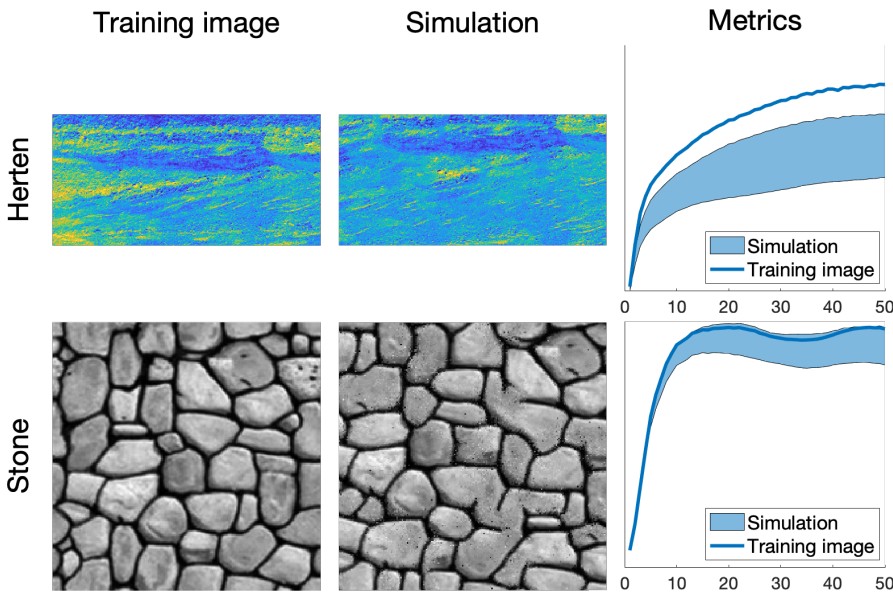


*Figure A 2 Examples of 2D simulations: the first 3 rows represent 3 variables of a single*
*simulation*





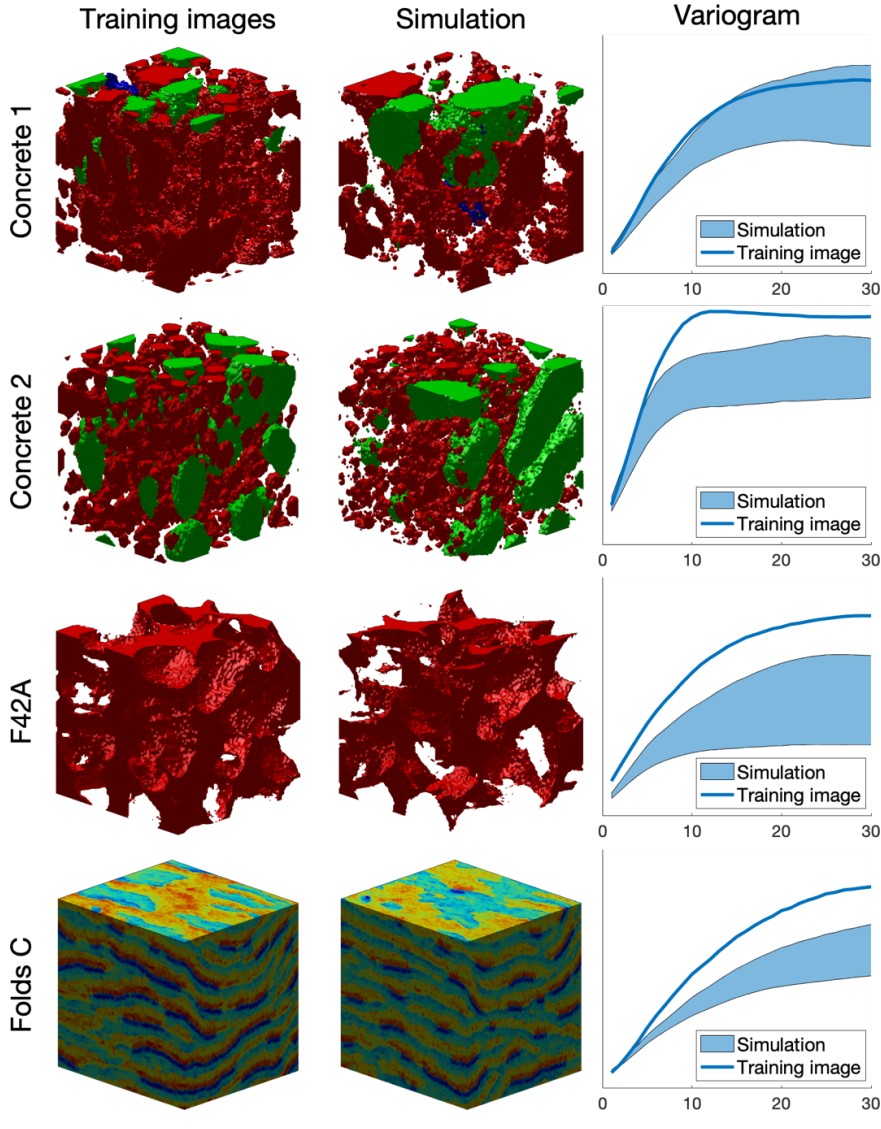

*Figure A 3 Examples of 3D simulation results*





| | Herten | Stone |
|---|---|---|
| Source | (Mariethoz and Caers, 2014) | (Mariethoz and Caers, 2014) |
| Size of the training image (px) | $716 \times 350$ | $200 \times 200$ |
| Size of the Simulation (px) | $716 \times 350$ | $200 \times 200$ |
| Computation time (s) | 1133 | 21 |
| $k$ | 1.2 | |
| $N$ | 80 | |

*Table A 3 Simulation parameters for Figure A 2. Times are specified for simulations without*
*parallelization.*

| | Concrete 1 | Concrete 2 | F42A | Folds continues |
|---|---|---|---|---|
| Source | (Meerschman et al., 2013) | (Meerschman et al., 2013) | Doi:10.6084/m9.fig share.1189259 | (Mariethoz and Caers, 2014) |
| Size of the training image (px) | $150 \times 150 \times 150$ | $100 \times 90 \times 80$ | $100 \times 100 \times 100$ | $180 \times 150 \times 120$ |
| Size of the simulation (px) | $100 \times 100 \times 100$ | $100 \times 100 \times 100$ | $100 \times 100 \times 100$ | $180 \times 150 \times 120$ |
| Computation time (s) | 11436 | 1416 | 1638 | 7637 |
| $k$ | 1.2 | | | |
| $N$ | 50 | | 125 | |

*Table A 4 Simulation parameters for Figure A 3. Times are specified for simulations without*
*parallelization.*



*Figure A 4 Complete sensitivity analysis, with one simulation for the two main parameters of QS*

*Figure A 5 Complete sensitivity analysis, with one simulation for each kernel with k=1.5 and N=40*



## 9. Acknowledgments

This research was funded by the Swiss National Science Foundation, grant number 200021_162882. Thanks to Intel for allowing us to conduct numerical experiments on their latest hardware using the AI DevCloud. Thanks to Luiz Gustavo Rasera for his comments, which greatly improved the manuscript; to Ehsan Baninajar for running his optimization method, which improved the reliability of the benchmarks; and to all the early users of QS for their useful feedback and their patience in waiting for this manuscript.

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
