# Peer review of "Quantile Sampling: a robust and simplified pixel-based"

_Geoscientific Model Development, 2019_

## Short Comment (SC1) · 16 Dec 2019

Dear authors,

in my role as Executive editor of GMD, I would like to bring to your attention our Editorial version 1.2:

https://www.geosci-model-dev.net/12/2215/2019/

This highlights some requirements of papers published in GMD, which is also available on the GMD website in the 'Manuscript Types' section:

http://www.geoscientific-model-development.net/submission/manuscript_types.html

[Figure]

In particular, please note that for your paper, the following requirement has not been met in the Discussions paper:

- "The main paper must give the model name and version number (or other unique identifier) in the title."

Please add a version number for the QS in the title upon your revised submission to GMD. Yours,

Astrid Kerkweg
* * *

---

## Referee Comment (RC1) · Ute Mueller (Referee) · 26 Dec 2019

The paper describes a new algorithm for multiple point simulation of continuous and discrete spatial variables. To start with a short review of the various types of MPS algorithms is provided, which distinguishes patching from pixel based approaches. The algorithm described here falls into the second category. Shortcomings of the method are discussed briefly, including the need for a threshold and sensitivity of the simulation quality to this threshold, but which can also lead to very long simulation times. In this paper the authors exploit a decomposition of the distance measures to apply FFT to speed up computation of mismatch maps with the aim to more quickly identify

candidate patterns in the training image, which may be complete of incomplete. The use of the FFT to compute the mismatch map is attractive in that it is fast to compute irrespective of dimension.

The mismatch map is calculated by computing for each pair (s, t) a dissimilarity measure where t belongs to the training image and s to the conditioning set. It is this dissimilarity measure which is then identified in terms of cross correlation. The authors provide a description of the metrics applied and a rewrite of the metrics in terms of cross correlations, and while the reader gets a general idea as to what is being calculated the derivation is patchy and somewhat sloppy in that summation indices are missing and critical steps are not described satisfactorily, such as the derivation of equation 9, which introduces cross correlations. Also, is it correct to assume that "l" is a grid operator? Once the mismatch map is computed, the k best matches are identified and a sample is drawn at random from this pool. The possibility of having non-integer values for k is touched upon, and allow unequal weighting of the first ceiling(k) candidates, with the first floor(k) candidates equally likely and the final candidate less likely (probability of being chose): 1-floor(k)/k) . The main advantage appears to lie in being able to choose between 2 instead of just one candidate (case of k between 1 and 2)

Simplifications and computational implementation details for speeding up the computation are discussed reasonably thoroughly and provide other practitioners with useful suggestions on how to potentially improve the efficiency of their own MPS algorithms. The proposed algorithm is benchmarked by means of standard sample data sets and a sensitivity analysis is provided demonstrating that QS performs well subject to the choice of a suitable kernel and that the quality of QS simulations is similar to that of DS simulations. It would have been interesting to see an exploration of kernels other than one of Gaussian type. Also, the metrics being used to assess the performance would benefit from going beyond variograms and connectivity (I acknowledge that the Euler characteristic was also used, but what good is it without a definition? Reference to another paper is all fine and well, but a definition and an explanation of what it measures would have been nice.) It would be really nice to see an evaluation in terms of a multipoint statistics.

Please amend all the formulae to ensure summation indices are clear, eg: Line 149: It is not clear over what is summed in equation 1. You clarify this to some extent below in lines 150 to 183, but I find this a little unsatisfying Line 174: The description preceding equation 2 talks about vectors, but the formula seems to be univariate. If you have c categories, is "a" a vector with c entries or simply one of the values from 1 to c if you label the categories in that manner?, It looks to me that "a" is simply a category . . . so looking at the equation, it would seem that it is equal to c, if "a" and "b" are distinct and equal to c-1 if they are equal, while the sum on the right is equal to 1 if "a" and "b" are equal and 0 else. There are also brackets missing in the middle expression (you should have \sum_{j \in C} (1-\delta_{aj}\delta_{bj}) Line 200: N(t) is not just a location but a neighbourhood? Please clarify Line 230: define the cross-correlation operator. Also, T_i has not been defined. You identify "*" with convolution and then apply the convolution theorem. Provide a derivation that this is true in an appendix. There are also some typos in the figure captions

---

## Referee Comment (RC2) · Thomas Mejer Hansen (Referee) · 27 Dec 2019

The authors present a novel multiple point statistical simulation algorithm that works for both discrete and continuous data, that scales well on parallel computing architectures, and that is available as open-source C++ code (G2S) with interfaces in Matlab, Python and R.

At the core of the method is the use of convolution to very efficiently compute to compute a mismatch, between a conditional event (consisting of the 'N' closest hard/simulated data) centered at all locations in the TI (except near the boundaries) (2.3) Then the authors suggest to simulate the current pixel based on a random se-

lection between the 'k' centered pixel values associated with the smallest mismatch
(2.4)

This leads to an algorithm with only two main 'tuning parameters'. The algorithm is in-itself novel and has obvious potential for used instead of some of the currently widely used MPS methods. The examples in the manuscript nicely describe the potential uses. In addition, the way the algorithm has been implemented should be applauded, as it is available as Open Source code that can be used with ease ranging from a case of "running on a single thread on a laptop in python/Matlab", to "running remote on a large cluster". This makes the code very versatile.

Therefore I find the manuscript highly appropriate for publication.

I have one major comment, that relate to the name of the algorithm and the way a pixel value is chosen based on 'k' smallest values of E/mismatch. The authors refer to these 'k' smallest values of E as a "quantile" and call the algorithm, for quantile sampling. This I do not understand and find a bit misleading. How can this represent a quantile? I think the term 'threshold' would be more fitting than 'quantile'.

The use of the term "quantile" suggests that the selection of the new pixel value is based of a probabilistic measure. Also, say 'k=18', and for a discrete case only 9 pixel values are associated with a mismatch of '0'. Why would one want to use the same probability (P=9/18) to select one of these, as opposed to one of the pixel values associated with a non-perfect match (P=9/10)? Or more extreme, say that pixel associated with the 18th best mismatch has a mismatch of 10 pixels. Why would one want to assign the same probability (1/18) to this, as to the pixels with a mismatch of 0? The use of the 'k'-'threshold' is convenient, but to me it makes the method less clear to describe in terms of the implied statistical assumptions. Some discussion on 'quantile' vs 'threshold' would be good.

Some comments to the text:

Line 150: Here 'a' and 'b' are referred to as "univariate pixel values". It seems 'a' and 'b' has a different meaning in line 174 (eqn 3)? Here they seem to represent vectors?

Line 185, Eqn 5: Please elaborate a bit on how this allows mixing discrete and continuous variables calculating the mismatch? It seems nontrivial to compute the mismatch between for example a velocity of 2.1 km/s and a "lithology of type A" to a velocity of 2.13 km/s and "lithology of type C"?

Figure 1: What do the red dots in the middle small figure?

Line 287: Please explain clearly what is meant by "verbatim copy". The term is used several places without a proper definition.

Line 338: Please explain "NUMA-aware" or provide a reference.

Line 392: What is meant by "..enables adaption of the parameterization..."?

Figure 5: Please help the reader here: is Qs with a kernel better than QS with no kernel? I am not sure what the figure tells us?

Line 399, Figure 6:Perhaps you could elaborate a little bit on "Euler characteristic" and whether it is a problem what Figure 6 shows?

Figure 8: I need some help appreciating how Figure 8 suggests that the use of alpha is useful?

Figure 9: Please show the 'dots' (the actual CPU time measurements) in the figures. Is it fair to say that the main limitation of the using QS is the size of the training image?

Lines 466-472. It is nice that one can choose to use many conditional point with not extra CPU costs. one could though argue that sometimes it is convenient in other MPS methods (SNESIM/IMPALA/DS) that the simulation becomes MUCH faster if one uses few conditioning data. If you would want to simulate with fewer conditioning data, QS would not lead to faster CPU time.. Just to say that the advantage you describe, could in a specific context, be seen as the opposite.
Some of the figures and tables in Appendix A should be excluded unless they are discussed and references in the text.
* * *

---

## Author Comment (AC1) · 4 Feb 2020

Reply to Reviewer 1 (Prof. Ute Mueller)

**The paper describes a new algorithm for multiple point simulation of continuous and**
**discrete spatial variables. To start with a short review of the various types of MPS algorithms**
**is provided, which distinguishes patching from pixel based approaches. The algorithm**
**described here falls into the second category. Shortcomings of the method are discussed**
**briefly, including the need for a threshold and sensitivity of the simulation quality to this**
**threshold, but which can also lead to very long simulation times. In this paper the authors**
**exploit a decomposition of the distance measures to apply FFT to speed up computation of**
**mismatch maps with the aim to more quickly identify candidate patterns in the training**
**image, which may be complete of incomplete. The use of the FFT to compute the mismatch**
**map is attractive in that it is fast to compute irrespective of dimension.**

We thank Prof. Ute Mueller for her feedback and interest in our work

**The mismatch map is calculated by computing for each pair (s, t) a dissimilarity measure**
**where t belongs to the training image and s to the conditioning set. It is this dissimilarity**
**measure which is then identified in terms of cross correlation. The authors provide a**
**description of the metrics applied and a rewrite of the metrics in terms of cross correlations,**
**and while the reader gets a general idea as to what is being calculated the derivation is**
**patchy and somewhat sloppy in that summation indices are missing and critical steps are**
**not described satisfactorily, such as the derivation of equation 9, which introduces cross**
**correlations.**

We will complete the notations by adding summation indices in all equations. The derivation
of equation 9 will be described in an appendix.

**Also, is it correct to assume that "l" is a grid operator?**

*l* represents lag vectors. Therefore, here it represents displacement on the grid. We will add
a clarification about it in the manuscript.

**Once the mismatch map is computed, the k best matches are identified and a sample is**
**drawn at random from this pool. The possibility of having non-integer values for k is touched**
**upon, and allow unequal weighting of the first ceiling(k) candidates, with the first floor(k)**
**candidates equally likely and the final candidate less likely (probability of being chose): 1-**
**floor(k)/k) . The main advantage appears to lie in being able to choose between 2 instead of**
**just one candidate (case of k between 1 and 2)**

We agree with the reviewer. Another advantage we see is that it provides an equivalence
between QS and the DS approach, allowing for benchmarks. In fact, DS with a threshold of t=0
and a scanning fraction of f=1/k can be seen as equivalent to QS. A discussion on this will be
added to the manuscript.

**Simplifications and computational implementation details for speeding up the computation**
**are discussed reasonably thoroughly and provide other practitioners with useful**
**suggestions on how to potentially improve the efficiency of their own MPS algorithms. The**
**proposed algorithm is benchmarked by means of standard sample data sets and a sensitivity**
**analysis is provided demonstrating that QS performs well subject to the choice of a suitable**

**41  kernel and that the quality of QS simulations is similar to that of DS simulations. It would**
**42  have been interesting to see an exploration of kernels other than one of Gaussian type.**

While we agree that a full exploration of kernel possibilities would be interesting, it will be the
subject of a future paper.
To be more precise about this point, preliminary experimentations on various kernel radial
designs have shown that it is not straightforward to define general guidelines for an optimal
kernel. Figure 1 below shows some of these preliminary results, where exhaustive kernel
parameter exploration is carried out and identifies areas of higher performance in terms of
reproducing variogram and connectivity function. While it is clear that some kernel functions
perform better than others, it seems that the results are highly specific to the type of patterns
to reproduce and should be further investigated.
Furthermore, tests using non-parametric kernels show a potential for future improvements.
However, substantial future research is still needed on this topic, which will be the object of a
future publication.

[Figure]

*Figure 1 Errors for different types of radial kernel based on the stone image.*

In addition, our formulation may have been unclear, in the sense that the kernel used in this
manuscript has an exponential form and not Gaussian (the notation $\|.\|_2$ denotes the $L_2$ norm
and not its squared form in equation 16) . We will clarify this point to make the paper easier
to read.

**63  Also, the metrics being used to assess the performance would benefit from going beyond**
**64  variograms and connectivity (I acknowledge that the Euler characteristic was also used, but**
**65  what good is it without a definition? Reference to another paper is all fine and well, but a**
**66  definition and an explanation of what it measures would have been nice.)**

As advised, we will add some brief explanation in the paper about the Euler characteristic and
connectivity metrics ~lines 385-391.

**69  It would be really nice to see an evaluation in terms of a multipoint statistics.**

To address this comment, we will carry out a validation of our realizations in terms of the
reproduction of higher-order statistics, using cumulants as a metric. Another possibility would
be the use of multiple-point histograms, but we refer not to use them because their
interpretation can be very difficult, and moreover are possible only for categorical variables.

**74** **Please amend all the formulae to ensure summation indices are clear, eg: Line 149: It is not**
**75** **clear over what is summed in equation 1.**

**76** Unfortunately, in Equation 1, line 149, it is impossible to know in advance the number of
**77** elements or the set for the summation. The description is really generic here and needs to be
**78** adapted for each required metric as shown for the $L_2$ and Hamming metrics. However, we will
**79** add and define proper ensemble for each summation to help the reader.

**80** **You clarify this to some extent below in lines 150 to 183, but I find this a little unsatisfying**
**81** **Line 174: The description preceding equation 2 talks about vectors, but the formula seems**
**82** **to be univariate.**

**83** We agree this is unclear as the "vector" in line 172, was referring to the origin of the Hamming-
**84** distance. We will rephrase this sentence to remove any confusion for the audience as
**85** following.

**86** **If you have c categories, is "a" a vector with c entries or simply one of the values from 1 to**
**87** **c if you label the categories in that manner?, It looks to me that "a" is simply a category ...**
**88** **so looking at the equation, it would seem that it is equal to c, if "a" and "b" are distinct and**
**89** **equal to c-1 if they are equal, while the sum on the right is equal to 1 if "a" and "b" are equal**
**90** **and 0 else. There are also brackets missing in the middle expression (you should have**
**91** **\sum_{j \in C} (1-\delta_{aj}\delta_{bj})**

**92** The description with categorical cases described by the reviewer is correct (and we don't need
**93** to number from 1 to c, and it is not the case in the implementation either). However, a mistake
**94** sneaked in, and we thank the reviewer for spotting the error of the equation 3. Indeed. It
**95** should be: $1 - \sum_{j\epsilon C}(\delta_{a,j}\cdot\delta_{b,j})$ and not $\sum_{j\epsilon C} 1 - (\delta_{a,j}\cdot\delta_{b,j})$. Therefore, now equation 3 is:

**96**
$$\epsilon_{L^0}(a,b) = 1 - \sum_{j\epsilon C}\delta_{a,j}\cdot\delta_{b,j} \propto \sum_{j\epsilon C}\delta_{a,j}\cdot\delta_{b,j}$$

**97** The linear transformation between both sides of the proportional symbol is y=ax+b, with a=-
**98** 1, and b=1.

**99** **Line 200: N(t) is not just a location but a neighbourhood?**

**100** $N(t)$ is indeed a neighborhood. We agree line 203 can be ambiguous, and we will rephrase it
**101** as follows: where $N_l(p)$ represents the neighbor value (or vector) at the position $p + l$, ($p$ can
**102** represent either $s$ or $t$)

**103** **Please clarify Line 230: define the cross-correlation operator. Also, T_i has not been defined.**
**104** **You identify "*" with convolution and then apply the convolution theorem. Provide a**
**105** **derivation that this is true in an appendix.**

**106** $\star$ represents the cross-correlation and therefore the "convolution theorem" is applied as
**107** follows: $\mathcal{F}(x \star y) = \overline{\mathcal{F}(x)} \circ \mathcal{F}(y)$, contrarily to a convolution $*$ where we get $\mathcal{F}(x * y) =$
**108** $\mathcal{F}(x) \circ \mathcal{F}(y)$. A clarification will be added and $T_i$ will be properly defined.

**109** **There are also some typos in the figure captions**

**110** Captions will be checked and corrected in consequence.
**111**

Response to Reviewer 2 (Prof. Thomas Mejer Hansen)

**The authors present a novel multiple point statistical simulation algorithm that works for**
**both discrete and continuous data, that scales well on parallel computing architectures, and**
**that is available as open-source C++ code (G2S) with interfaces in Matlab, Python and R.**
**At the core of the method is the use of convolution to very efficiently compute to compute**
**a mismatch, between a conditional event (consisting of the 'N' closest hard/simulated data)**
**centered at all locations in the TI (except near the boundaries) (2.3) Then the authors**
**suggest to simulate the current pixel based on a random selection between the 'k' centered**
**pixel values associated with the smallest mismatch (2.4)**
**This leads to an algorithm with only two main 'tuning parameters'. The algorithm is in-itself**
**novel and has obvious potential for used instead of some of the currently widely used MPS**
**methods. The examples in the manuscript nicely describe the potential uses. In addition,**
**the way the algorithm has been implemented should be applauded, as it is available as Open**
**Source code that can be used with ease ranging from a case of "running on a single thread**
**on a laptop in python/Matlab", to "running remote on a large cluster". This makes the code**
**very versatile.**
**Therefore I find the manuscript highly appropriate for publication.**

Thanks a lot for the positive feedback!

**I have one major comment, that relate to the name of the algorithm and the way a pixel**
**value is chosen based on 'k' smallest values of E/mismatch. The authors refer to these 'k'**
**smallest values of E as a "quantile" and call the algorithm, for quantile sampling. This I do**
**not understand and find a bit misleading. How can this represent a quantile? I think the term**
**'threshold' would be more fitting than 'quantile'.**

The question of the algorithm name is something that has been extensively discussed
between authors. We believe that the use of the term "threshold" would bring confusion with
the Direct Sampling algorithm, which uses a threshold in the error.
One name that was originally discussed for our algorithm is "Quick Sampling". According to
the justified comment by the reviewer, we propose to use this name as a replacement, as it
allows keeping the acronym "QS" with which users are familiar.

**The use of the term "quantile" suggests that the selection of the new pixel value is based of**
**a probabilistic measure. Also, say 'k=18', and for a discrete case only 9 pixel values are**
**associated with a mismatch of '0'. Why would one want to use the same probability**
**(P=9/18) to select one of these, as opposed to one of the pixel values associated with a non-**
**perfect match (P=9/10)? Or more extreme, say that pixel associated with the 18th best**
**mismatch has a mismatch of 10 pixels. Why would one want to assign the same probability**
**(1/18) to this, as to the pixels with a mismatch of 0? The use of the 'k'-'threshold' is**
**convenient, but to me it makes the method less clear to describe in terms of the implied**
**statistical assumptions. Some discussion on 'quantile' vs 'threshold' would be good.**

This hypothesis is similar to the use of a distance threshold in DS and should indeed be
discussed. k in QS is statistically similar to DS with a threshold at 0 and a fraction f=1/k (this is
one of the reasons for using decimal values for k) under the hypothesis of stationarity.
This equivalence will be discussed in section 2.4, to allow readers to get a better feeling about
the relation between QS and DS, and the utility of non-integer values of k.

The question of the reviewer could be turned around "why to not using the best candidate
(k=1)?". The main answer is to limit verbatim copy, because the random selection between
candidates with similar mismatch (algorithm presented in Appendix A.1) significantly limits
this problem in cases of a training image with replicated patterns. The problem remains for
other images, and especially continuous variable images where there exist often few
replicated patterns.

**Some comments to the text:**
**Line 150: Here 'a' and 'b' are referred to as "univariate pixel values". It seems 'a' and 'b' has**
**a different meaning in line 174 (eqn 3)? Here they seem to represent vectors?**

"a" and "b" represent each time one possible class. This section will be corrected and clarified
to remove this ambiguity, also according to the comments by reviewer 1.

**Line 185, Eqn 5: Please elaborate a bit on how this allows mixing discrete and continuous**
**variables calculating the mismatch? It seems nontrivial to compute the mismatch between**
**for example a velocity of 2.1 km/s and a "lithology of type A" to a velocity of 2.13 km/s and**
**"lithology of type C"?**

The task of combining continuous and categorical variables is indeed challenging, and has
been so for all MPS approaches. From the literature and practical use of the software, we
know that this problem is general to most MPS methods and that many strategies can be used.
One can use a different distance threshold for each variable (as done in the DEESSE
implementation), or a linear combination of the normalized errors (as done in the DS
implementation). Here we use the second approach, taking advantage of the linearity of the
Fourier transform. If the relative importance can be set in the "$f_i$" or "$g_i$" functions in equation
1, it is computationally advantageous to use the kernel weights such as to have standard
functions for each metric.
If the task of setting such variable-dependent parameters is complex, one can use the results
of recent research to identify the optimal parameterization using stochastic optimization
approaches, as in (Baninajar et al.), which can and has been applied on QS. This discussion will
be added in the revised manuscript.
Baninajar, E., Sharghi, Y. & Mariethoz, G. MPS-APO: a rapid and automatic parameter
optimizer for multiple-point geostatistics. *Stoch Environ Res Risk Assess* 33, 1969–1989
(2019). https://doi.org/10.1007/s00477-019-01742-7

**Figure 1: What do the red dots in the middle small figure?**

We will add to the caption that the pink pixels represent missing data.

**Line 287: Please explain clearly what is meant by "verbatim copy". The term is used several**
**places without a proper definition.**

A short explanation and a reference will be added to clarify the meaning of verbatim copy.

**Line 338: Please explain "NUMA-aware" or provide a reference.**

NUMA stands for "Non-Uniform Memory Access" and refers to memory communication
between many CPU sockets (such as bi-Xeon). This connection has a limited bandwidth and
therefore minimizing the communication on it can significantly increase the speed of the algorithm on such architectures. This can have a huge impact when running on workstations
or clusters computers. A reference about this will be added.

**Line 392: What is meant by "..enables adaption of the parameterization. . ."?**

Here we mean that it allows fine-tuning the parameterization. It will be clarified in the revised
manuscript.

**Figure 5: Please help the reader here: is Qs with a kernel better than QS with no kernel? I**
**am not sure what the figure tells us?**

We agree that these figures are currently not very well explained and that the text describing
them can be improved. We will do this by merging figures 5 and 6 and adding comments.
These figures mean to convey the message that the patterns are well reproduced in all
approaches, however QS presents a better reproduction of the metrics

**Line 399, Figure 6: Perhaps you could elaborate a little bit on "Euler characteristic" and**
**whether it is a problem what Figure 6 shows?**

It will be added, also according to the comment by reviewer 1, by extending the description
of the metrics.

**Figure 8: I need some help appreciating how Figure 8 suggests that the use of alpha is useful?**

We agree that this figure does not illustrate very well the use of the alpha parameter.
However, over many tests, and as confirmed by feedback of early users, this parameter does
allow a fine tuning of the simulation and is therefore an interesting tool, especially for
conditional simulations with an exhaustively informed covariable. The goal here is to make
the reader aware of this possibility. We will therefore change figure 8 to better show the
sensitivity to alpha, possibly using a different case study.

**Figure 9: Please show the 'dots' (the actual CPU time measurements) in the figures. Is it fair**
**to say that the main limitation of the using QS is the size of the training image?**

The "dots" will be added. We agree that the main limitation of QS is the TI size because its
relation to computing time evolves in $O(n.\ln(n))$ due to the FFT computation. If solutions such
as window convolution exist (often used in audio processing), in our tests the improvements
are only noticeable for huge TIs. While such approaches do bring an improvement and tend
to reduce the memory footprint, they also add significant complexity to the algorithm for a
minor gain. Over the last decade, convolution techniques have been substantially improved,
driven by the needs of Convolutional Neural Networks, but are often applied to small matrices
(e.g. 3x3, 5x5, or 7x7). Other solutions are available to increase the speed of the convolution
such as GPUs or FPGAs that we are still investigating.
Note that a dedicated CUDA implementation of QS is available in our repository, but it is still
work in progress and at this stage we prefer not to include a detailed description of it in the
paper.

**Lines 466-472. It is nice that one can choose to use many conditional point with not extra**
**CPU costs. one could though argue that sometimes it is convenient in other MPS methods**
**(SNESIM/IMPALA/DS) that the simulation becomes MUCH faster if one uses few**
**conditioning data. If you would want to simulate with fewer conditioning data, QS would**

**not lead to faster CPU time.. Just to say that the advantage you describe, could in a specific context, be seen as the opposite.**

We completely agree with this point. However, the current trend in the field is to reduce the simulation quality in order to gain in time or memory space. The point here was to explain that whatever the parameterization (and quality), the computation time is identical. Therefore, it is better to choose parameters yielding a good quality simulation.

This is an important discussion point: QS will be fast with a small image, whereas for (SNESIM/IMPALA) it is only partially true because of the overhead related to the creation of the list/tree. Similarly, QS in insensitive to the complexity of the training image (number of patterns available), whereas SNESIM/IMPALA/DS are highly sensitive to it. QS is therefore more adapted to TIs with complex features and few repetitions. We however agree that SNESIM/IMPALA/DS will simulate significantly faster a simple and repetitive TI than QS, but in such cases computation time is generally not a critical issue anyway.

A discussion on these questions will be included in the revised manuscript.

**Some of the figures and tables in Appendix A should be excluded unless they are discussed and references in the text.**

We will fix the missing references.

Response to Short Comment 1 (Executive editor of GMD Astrid Kerkweg)

**Please add a version number for the QS in the title upon your revised submission to GMD.**

As suggested we will add the adapted versioning identifier in the title of the manuscript.

---

## Author Response (AR1)

**Submission of revised version of manuscript titled "QuickSampling v1.0: a robust and simplified pixel-based multiple-point simulation approach"**

We would like to thank Ute Mueller and Thomas Mejer Hansen for their valuable feedbacks. Their inputs led to significant improvements in the manuscript.

We have incorporated the suggestions made by the reviewers in the revised manuscript. We have added cumulants as multiple points statistics, and significantly improve the discussion (and figures) associated with the kernel to provide the readers a better understanding of this new parameter's benefits.

Furthermore, due to reviewers comments and external feedbacks we decided to rename the algorithm from Quantile Sampling to QuickSampling in order to prevent ambiguity.

We hope that with these additions and all modifications discussed below, the revised manuscript is now ready for publication in Geoscientific Model Development.

Best regards,
Mathieu Gravey

**Reply to Reviewer 1 (Prof. Ute Mueller)**

**The paper describes a new algorithm for multiple point simulation of continuous and discrete spatial variables. To start with a short review of the various types of MPS algorithms is provided, which distinguishes patching from pixel based approaches. The algorithm described here falls into the second category. Shortcomings of the method are discussed briefly, including the need for a threshold and sensitivity of the simulation quality to this threshold, but which can also lead to very long simulation times. In this paper the authors exploit a decomposition of the distance measures to apply FFT to speed up computation of mismatch maps with the aim to more quickly identify candidate patterns in the training image, which may be complete of incomplete. The use of the FFT to compute the mismatch map is attractive in that it is fast to compute irrespective of dimension.**

We thank Prof. Ute Mueller for her feedback and interest in our work.

**The mismatch map is calculated by computing for each pair (s, t) a dissimilarity measure where t belongs to the training image and s to the conditioning set. It is this dissimilarity measure which is then identified in terms of cross correlation. The authors provide a description of the metrics applied and a rewrite of the metrics in terms of cross correlations, and while the reader gets a general idea as to what is being calculated the derivation is patchy and somewhat sloppy in that summation indices are missing and critical steps are not described satisfactorily, such as the derivation of equation 9, which introduces cross correlations.**

We improved the overall readability by introducing "sets" in summation. Equation 8 was extended to add the reordering of the summation (Line 236), and the definition of cross-collation was added between equation 8 and 9 (Line 238) to enhance readability of the derivation.

**Also, is it correct to assume that "l" is a grid operator?**

*l* was properly defined as a relative displacement (Line 211).

**Once the mismatch map is computed, the k best matches are identified and a sample is drawn at random from this pool. The possibility of having non-integer values for k is touched upon, and allow unequal weighting of the first ceiling(k) candidates, with the first floor(k) candidates equally likely and the final candidate less likely (probability of being chose): 1-floor(k)/k . The main advantage appears to lie in being able to choose between 2 instead of just one candidate (case of k between 1 and 2)**

The interest of partial k, in particular the relation with Direct Sampling (DS) was further developed at Line 304 -308.

**Simplifications and computational implementation details for speeding up the computation are discussed reasonably thoroughly and provide other practitioners with useful suggestions on how to potentially improve the efficiency of their own MPS algorithms. The proposed algorithm is benchmarked by means of standard sample data sets and a sensitivity analysis is provided demonstrating that QS performs well subject to the choice of a suitable kernel and that the quality of QS simulations is similar to that of DS simulations. It would have been interesting to see an exploration of kernels other than one of Gaussian type.**

While we agree that a full exploration of kernel possibilities would be interesting, it will be the
subject of a future paper.
To be more precise about this point, preliminary experimentations on various kernel radial
designs have shown that it is not straightforward to define general guidelines for an optimal
kernel. Figure 1 below shows some of these preliminary results, where exhaustive kernel
parameter exploration is carried out and identifies areas of higher performance in terms of
reproducing variogram and connectivity function. While it is clear that some kernel functions
perform better than others, it seems that the results are highly specific to the type of patterns
to reproduce and should be further investigated.
Furthermore, tests using non-parametric kernels show a potential for future improvements.
However, substantial future research is still needed on this topic, which will be the object of a
future publication.

[Figure]

*Figure 1 Errors for different types of radial kernel based on the stone image.*

In addition, with improved Figure 8 by adding the Euler characteristics, that allows readers to
get a better idea of the potential and the interest of this kernels.
Because our formulation may have been unclear, in the sense that the example kernel used
in this manuscript has an exponential form and not Gaussian. Therefore, we clarified this point
at Line 423 by specifying that $\|.\|_2$ represents the Euclidean distance.

**Also, the metrics being used to assess the performance would benefit from going beyond**
**variograms and connectivity (I acknowledge that the Euler characteristic was also used, but**
**what good is it without a definition? Reference to another paper is all fine and well, but a**
**definition and an explanation of what it measures would have been nice.)**
**It would be really nice to see an evaluation in terms of a multipoint statistics.**

We provided some brief description of each metrics (Line 407 – 412). We carried out a
validation for our realizations in terms of the reproduction of higher-order statistics, using
cumulants as a metric in Figure 6.

**Please amend all the formulae to ensure summation indices are clear, eg: Line 149: It is not**
**clear over what is summed in equation 1.**

Unfortunately, in Equation 1, line 150, it is impossible to know in advance the number of
elements or the set for the summation. The description is really generic here and needs to be

| 97 | adapted for each required metric as shown for the $L_2$ and Hamming metrics. However, we |
| 98 | add and define proper ensemble $\mathcal{J}$ for each summation to help the reader. |

**You clarify this to some extent below in lines 150 to 183, but I find this a little unsatisfying Line 174: The description preceding equation 2 talks about vectors, but the formula seems to be univariate.**

We agree this is unclear as the "vector" in line 173, was referring to the origin of the Hamming-distance. We rephrase this sentence to remove any confusion for the audience as following: "The Hamming distance measures the dissimilarity between two lists by counting the number of elements that have different categories (Hamming, 1950)".

**If you have c categories, is "a" a vector with c entries or simply one of the values from 1 to c if you label the categories in that manner?, It looks to me that "a" is simply a category ... so looking at the equation, it would seem that it is equal to c, if "a" and "b" are distinct and equal to c-1 if they are equal, while the sum on the right is equal to 1 if "a" and "b" are equal and 0 else. There are also brackets missing in the middle expression (you should have \sum_{j \in C} (1-\delta_{aj}\delta_{bj})**

The description with categorical cases mentioned by the reviewer is correct (and we don't need to number from 1 to c, and it is not the case in the implementation either). We thank the reviewer for spotting the error of the equation 3. Indeed, It should be: $1 - \sum_{j \in \mathcal{C}} (\delta_{a,j} \cdot \delta_{b,j})$ and not $\sum_{j \in \mathcal{C}} 1 - (\delta_{a,j} \cdot \delta_{b,j})$. Therefore, now equation 3 is:

$$\epsilon_{L^0}(a,b) = 1 - \sum_{j \in \mathcal{C}} \delta_{a,j} \cdot \delta_{b,j} \propto \sum_{j \in \mathcal{C}} \delta_{a,j} \cdot \delta_{b,j}$$

The linear transformation between both sides of the proportional symbol is y=ax+b, with a=-1, and b=1. The manuscript is updated accordingly (Line 179).

**Line 200: N(t) is not just a location but a neighbourhood?**

$N(t)$ is indeed a neighborhood. We clarify this point in Line 210-212.

**Please clarify Line 230: define the cross-correlation operator. Also, T_i has not been defined. You identify "*" with convolution and then apply the convolution theorem. Provide a derivation that this is true in an appendix.**

$\star$ represents the cross-correlation and therefore the "convolution theorem" is applied as follows: $\mathcal{F}(x \star y) = \overline{\mathcal{F}(x)} \circ \mathcal{F}(y)$. This is contrarily to a convolution $*$ where we get $\mathcal{F}(x * y) = \mathcal{F}(x) \circ \mathcal{F}(y)$. Some references and a proper derivation of this property were added in the Annex Line 684. Furthermore, $T_i$ is properly defined in Line 244.

**There are also some typos in the figure captions**

Captions were checked and corrected in consequence.

Response to Reviewer 2 (Prof. Thomas Mejer Hansen)

**The authors present a novel multiple point statistical simulation algorithm that works for**
**both discrete and continuous data, that scales well on parallel computing architectures, and**
**that is available as open-source C++ code (G2S) with interfaces in Matlab, Python and R.**
**At the core of the method is the use of convolution to very efficiently compute to compute**
**a mismatch, between a conditional event (consisting of the 'N' closest hard/simulated data)**
**centered at all locations in the TI (except near the boundaries) (2.3) Then the authors**
**suggest to simulate the current pixel based on a random selection between the 'k' centered**
**pixel values associated with the smallest mismatch (2.4)**
**This leads to an algorithm with only two main 'tuning parameters'. The algorithm is in-itself**
**novel and has obvious potential for used instead of some of the currently widely used MPS**
**methods. The examples in the manuscript nicely describe the potential uses. In addition,**
**the way the algorithm has been implemented should be applauded, as it is available as Open**
**Source code that can be used with ease ranging from a case of "running on a single thread**
**on a laptop in python/Matlab", to "running remote on a large cluster". This makes the code**
**very versatile.**
**Therefore I find the manuscript highly appropriate for publication.**

Thanks a lot for the positive feedback!

**I have one major comment, that relate to the name of the algorithm and the way a pixel**
**value is chosen based on 'k' smallest values of E/mismatch. The authors refer to these 'k'**
**smallest values of E as a "quantile" and call the algorithm, for quantile sampling. This I do**
**not understand and find a bit misleading. How can this represent a quantile? I think the term**
**'threshold' would be more fitting than 'quantile'.**

The question of the algorithm name is something that has been extensively discussed
between authors. We believe that the use of the term "threshold" would bring confusion with
the Direct Sampling algorithm, which uses a threshold in the error.
One name that was originally discussed for our algorithm is "Quick Sampling". According to
the justified comment by the reviewer, we propose to use this name as a replacement, as it
allows keeping the acronym "QS" with which users are already familiar.

**The use of the term "quantile" suggests that the selection of the new pixel value is based of**
**a probabilistic measure. Also, say 'k=18', and for a discrete case only 9 pixel values are**
**associated with a mismatch of '0'. Why would one want to use the same probability**
**(P=9/18) to select one of these, as opposed to one of the pixel values associated with a non-**
**perfect match (P=9/10)? Or more extreme, say that pixel associated with the 18th best**
**mismatch has a mismatch of 10 pixels. Why would one want to assign the same probability**
**(1/18) to this, as to the pixels with a mismatch of 0? The use of the 'k'-'threshold' is**
**convenient, but to me it makes the method less clear to describe in terms of the implied**
**statistical assumptions. Some discussion on 'quantile' vs 'threshold' would be good.**

This hypothesis is similar to the use of a distance threshold in DS and should indeed be
discussed. k in QS is statistically similar to DS with a threshold at 0 and a fraction f=1/k (this is
one of the reasons for using decimal values for k) under the hypothesis of stationarity.
This equivalence was added in section 2.4, to allow readers to get a better understanding
about the relation between QS and DS, and the utility of non-integer values of k.

The question of the reviewer could be turned around "why to not using the best candidate
(k=1)?". The main answer is to limit verbatim copy, because the random selection between
candidates with similar mismatch (algorithm presented in Appendix A.1) significantly limits
this problem in cases of a training image with replicated patterns. The problem remains for
other images, and especially continuous variable images where often few replicated patterns
exist.

**Some comments to the text:**
**Line 150: Here 'a' and 'b' are referred to as "univariate pixel values". It seems 'a' and 'b' has**
**a different meaning in line 174 (eqn 3)? Here they seem to represent vectors?**

"a" and "b" represent each time one possible class. This section was corrected and clarified to
remove this ambiguity, also according to the comments by reviewer 1. In addition, in order to
help readers to better understand how this metric works, a short example was added in Line
176.

**Line 185, Eqn 5: Please elaborate a bit on how this allows mixing discrete and continuous**
**variables calculating the mismatch? It seems nontrivial to compute the mismatch between**
**for example a velocity of 2.1 km/s and a "lithology of type A" to a velocity of 2.13 km/s and**
**"lithology of type C"?**

The task of combining continuous and categorical variables is indeed challenging, and has
been so for all MPS approaches. From the literature and practical use of the software, we
know that this problem is general to most MPS methods and that many strategies can be used.
One can use a different distance threshold for each variable (as done in the DEESSE
implementation), or a linear combination of the normalized errors (as done in the DS
implementation). Here we use the second approach, taking advantage of the linearity of the
Fourier transform. If the relative importance can be set in the "$f_i$" or "$g_i$" functions in equation
1, it is computationally advantageous to use the kernel weights such as to have standard
functions for each metric.
If the task of setting such variable-dependent parameters is complex, one can use the results
of recent research to identify the optimal parameterization using stochastic optimization
approaches, as in (Baninajar et al.), which can and has been applied on QS. This discussion was
added in the revised manuscript Line 515-523.
Baninajar, E., Sharghi, Y. & Mariethoz, G. MPS-APO: a rapid and automatic parameter
optimizer for multiple-point geostatistics. *Stoch Environ Res Risk Assess* 33, 1969–1989
(2019). https://doi.org/10.1007/s00477-019-01742-7

**Figure 1: What do the red dots in the middle small figure?**

Red/Purple dots represent missing and unusable (border effect) data. The caption of Figure 1
was updated accordingly.

**Line 287: Please explain clearly what is meant by "verbatim copy". The term is used several**
**places without a proper definition.**

A short explanation and a reference were added in the introduction (Line 64-66) to clarify the
meaning of verbatim copy:

Verbatim copy (Mariethoz and Caers, 2014) refers to the phenomenon whereby a neighbor of
a pixel in the simulation is the neighbor in the training image. This results in large parts of the
simulation that are identical to the training image.

**Line 338: Please explain "NUMA-aware" or provide a reference.**

NUMA stands for "Non-Uniform Memory Access" and refers to memory communication
between many CPU sockets (such as bi-Xeon). This connection has a limited bandwidth and
therefore minimizing the communication on it can significantly increase the speed of the
algorithm on such architectures. This can have a huge impact when running on workstations
or clusters computers. The manuscript is updated with a reference.

**Line 392: What is meant by "..enables adaption of the parameterization. . ."?**

Here we meant that it allows for fine-tuning the parameterization. Line 419 was updated in
accordingly.

**Figure 5: Please help the reader here: is Qs with a kernel better than QS with no kernel? I**
**am not sure what the figure tells us?**

The Figure 5 simply show that the results have similar quality and variability DS, which is
considered a reference in the field. The benefits of Kernel is further detailed at Line 450-452
and in Figure 8

**Line 399, Figure 6: Perhaps you could elaborate a little bit on "Euler characteristic" and**
**whether it is a problem what Figure 6 shows?**

It is added to the manuscript, according to the comments by reviewers. This is further
discussed by extending the description of the metrics in Line 409-411 and in particular with its
interpretation in Line 450-452.

**Figure 8: I need some help appreciating how Figure 8 suggests that the use of alpha is useful?**

We agree that this figure does not illustrate very well the use of the alpha parameter.
However, over many tests, and as confirmed by feedback of early users, this parameter does
allow a fine tuning of the simulation and is therefore an interesting tool, especially for
conditional simulations with an exhaustively informed covariable. The goal here is to make
the reader aware of this possibility. Therefore, we changed figure 8 to better show the
sensitivity in particular by adding Euler characteristics curve for each case and improving the
discussion.

**Figure 9: Please show the 'dots' (the actual CPU time measurements) in the figures. Is it fair**
**to say that the main limitation of the using QS is the size of the training image?**

The "dots" are added to the figure. We agree that the main limitation of QS is the TI size
because its relation to computing time evolves in $O(n.\ln(n))$ due to the FFT computation. Even
with the use of solutions such as window convolution (often used in audio processing), in our
tests the improvements are only noticeable for huge TIs. While such approaches do bring an
improvement and tend to reduce the memory footprint, they also add significant complexity
to the algorithm for a minor gain. Over the last decade, convolution techniques have been
substantially improved, driven by the needs of Convolutional Neural Networks, but are often
applied to small matrices (e.g. 3x3, 5x5, or 7x7). Other solutions are available to increase the
speed of the convolution such as GPUs or FPGAs that we are still investigating.

Note that a dedicated CUDA implementation of QS is available in our repository, but there is
still work in progress and at this stage we prefer not to include a detailed description of it in
the paper.
A short discussion about this point is added in Line 534-537.

**Lines 466-472. It is nice that one can choose to use many conditional point with not extra**
**CPU costs. one could though argue that sometimes it is convenient in other MPS methods**
**(SNESIM/IMPALA/DS) that the simulation becomes MUCH faster if one uses few**
**conditioning data. If you would want to simulate with fewer conditioning data, QS would**
**not lead to faster CPU time.. Just to say that the advantage you describe, could in a specific**
**context, be seen as the opposite.**

We completely agree with this point. However, many researchers in the field tend to reduce
the simulation quality in order to gain in time or memory space. The point here was to explain
that whatever the parameterization (and quality), the computation time is identical.
Therefore, it is better to choose parameters yielding a good quality simulation.
This is an important discussion point: QS will be fast with a small image, whereas for
(SNESIM/IMPALA) it is only partially true because of the overhead related to the creation of
the list/tree. Similarly, QS is insensitive to the complexity of the training image (number of
patterns available), whereas SNESIM/IMPALA/DS are highly sensitive to it. QS is therefore
more adapted to TIs with complex features and few repetitions. We however agree that
SNESIM/IMPALA/DS will simulate significantly faster a simple and repetitive TI than QS, but in
such cases computation time is generally not a critical issue anyway.
This was already discussed, but we updated it to add the relation with IMPALA and SNESIM.

**Some of the figures and tables in Appendix A should be excluded unless they are discussed**
**and references in the text.**

Missing references are added in the text.

Response to Short Comment 1 (Executive editor of GMD Astrid Kerkweg)

**Please add a version number for the QS in the title upon your revised submission to GMD.**

As suggested we add the versioning identifier in the title of the manuscript.

[revised manuscript text omitted]